# DEVA: Fine-tuning Multimodal Large Language Models for Visual Perception Tasks

## Abstract

Fine-tuning large language models (LLMs) using reinforcement learning (RL) objectives has gained traction, especially in scenarios where labeled data is limited. Building on its success in the language domain, recent efforts have extended RL-based fine-tuning to multimodal tasks. Visual-RFT, for instance, applied Group Relative Policy Optimization (GRPO) to fine-tune multimodal LLMs (MLLMs) across various visual perception benchmarks, achieving notable improvements over standard supervised fine-tuning (SFT). However, its scope was limited by a narrow evaluation of RL adaptation strategies. In this work, we expand the landscape by introducing new RL-based baselines on the same benchmarks and conducting a deeper analysis of GRPO's training dynamics. We identify key limitations—such as reduced generation diversity, constrained policy exploration, and suboptimal reward formulation and aggregation. To address these, we propose DEVA: a framework that enhances Diversity via a flow-based training objective, encourages broader policy Exploration through global entropic regularization, and leverages alignment Volume as a non-verifiable reward combined with harmonic Aggregation. Applied to GRPO and other RL methods, DEVA delivers consistent gains in both quantitative (+5 to +13 points) and qualitative metrics. We further provide visualizations, ablations, and analyses to unpack the contributions of each component in our framework.

## 1 Introduction

Recent progress in large reasoning models (LRMs) (Jaech et al., 2024) has demonstrated substantial gains on complex reasoning tasks, including mathematics and program synthesis. These improvements are often attributed to enhanced inference-time strategies, where models perform intermediate reasoning before producing final outputs. A notable example is o1 from OpenAI[1], which reportedly achieves significant performance boosts by fine-tuning on small, domain-specific datasets. While o1 remains closed-source, recent open-source efforts such as DeepSeek (Guo et al., 2025) highlights the effectiveness of incorporating verifiable rewards (Lambert et al., 2024; Guo et al., 2025; Team et al., 2025) during reinforcement learning (RL). In this paradigm, reward signals for algorithms like Group Relative Policy Optimization (GRPO) (Shao et al., 2024) are computed using deterministic rules derived from ground-truth solutions, offering more reliable and computationally efficient alternative to preference-based reward models (Ouyang et al., 2022b; Liu et al., 2024b; Zang et al., 2025) or process reward models (Cui et al., 2025; Wang et al., 2025a) that provide fine-grained feedback.

Reinforcement Learning (RL) offers distinct advantages over Supervised Fine-Tuning (SFT) for model adaptation. Prior work by Chu et al. (2025) demonstrates that RL-based fine-tuning promotes generalization, whereas SFT tends to encourage memorization. Generalization is particularly critical when working with small-scale datasets, as in our setting, to mitigate overfitting. In essence, SFT relies on "ground truth" responses and typically requires large-scale data, employing next-token prediction as the fine-tuning objective in multimodal large language models (MLLMs). Conversely, RL-based fine-tuning iteratively updates the model by leveraging feedback signals derived from its own responses across multiple episodes. This paradigm has been widely adopted in domains such as scientific reasoning and code generation. In this work, we focus on fine-tuning for visual perception tasks using the ViRFT benchmark (Liu et al., 2025b). ViRFT introduces rule-based,

---

[1]https://openai.com/form/rft-research-program

verifiable reward functions for GRPO, augmented with reasoning steps to refine model outputs, where rewards are aligned with task-specific metrics (e.g., IoU for object detection). This framework achieves substantial improvements over standard SFT, underscoring the efficacy of RL in enhancing visual perception and reasoning capabilities.

In this work, we introduce a **fine-tuning** strategy designed to improve upon RL-based adaptation. Unlike **test-time** scaling approaches, which are orthogonal to our focus, we concentrate exclusively on fine-tuning methods. Specifically, we target key limitations of GRPO and related RL variants for MLLM adaptation. A primary challenge lies in lack of diversity in rule-based reward functions, where outputs within a group often receive nearly identical rewards. This results in negligible advantage estimates, leading to vanishing policy gradients and ineffective policy updates. To address this, we

| Method | Avg. Std (↑) |
|---|---|
| GRPO | 0.234 |
| + GFlowNet Loss | **0.262** |
| DAPO | 0.201 |
| + GFlowNet Loss | **0.240** |
| BNPO | 0.193 |
| + GFlowNet Loss | **0.221** |

Figure 1: Effect of GFlowNet Loss on Reward Avg. Std. for the LISA dataset.

propose incorporating GFlowNet-based loss (Bengio et al., 2023) as an auxiliary training objective alongside the RL objective. GFlowNet training objectives have previously been employed to generate diverse reasoning trajectories in scientific domains (Yu et al., 2024a; Kwon et al., 2024). In contrast, our goal is to enhance reward diversity within groups, thereby providing stronger learning signals. *To the best of our knowledge, this is the first work to integrate GFlowNet loss with GRPO.* As illustrated in Fig. 1, introducing GFlowNet objective significantly increases reward diversity (measured by average standard deviation), which in turn translates into improved visual recognition performance, as demonstrated in our experiments.

GRPO and related RL algorithms incorporate an additional regularization term to ensure training stability, typically by penalizing the KL divergence between the per-token probability distributions of the policy model and a reference model. However, this localized, token-level regularization constrains the policy model's exploration capability, which can negatively impact visual perception recognition performance. To address this limitation, we propose a global entropic divergence regularization term. Specifically, we compute the entropy of the output distribution for all tokens separately for both the policy and reference models, and then measure the divergence between these entropy values. This global regularization encourages broader exploration of the policy space, which is particularly important for visual perception tasks, as it facilitates adapting a general-purpose MLLM to a specialized set of vision-oriented tasks.

In ViRFT, rewards are verifiable and derived from ground-truth annotations used during fine-tuning. For correct predictions, an accuracy-based reward is employed. However, no verifiable reward can be assigned to intermediate reasoning traces due to the absence of ground-truth reasoning steps. To address this gap, we introduce an alignment reward that encourages consistency between the reasoning trace, the input image, and the query. As illustrated in Fig. 2, pairwise alignment combined with aggregation introduces heterogeneous reward dynamics, where individual reward pairs peak and decline at different iterations. Empirically, this misalignment leads to sub-optimal visual perception performance. To overcome this limitation, we propose a reward that is

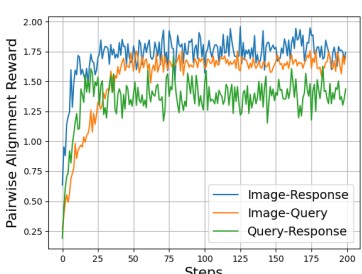

Figure 2: Evolution of Pairwise Alignment Rewards on the LISA dataset.

inversely proportional to the hyper-volume spanned by the three modalities, ensuring unified alignment across all components. Furthermore, we investigate strategies for aggregating multiple rewards. Our analysis reveals that simple arithmetic summation is sub-optimal, as dominant rewards can overshadow others, degrading performance. In contrast, harmonic aggregation proves more effective by enforcing simultaneous improvement across all rewards. We term our framework **DEVA**, reflecting its focus on enhancing **D**iversity and **E**xploration while incorporating alignment **V**olume as a non-verifiable reward that can be **A**ggregated through principled mechanisms.

In summary, our key contributions are as follows:

1. We introduce a GFlowNet-based objective to improve reward diversity among sampled responses, yielding stronger policy gradients and more effective training.

2. We enhance policy exploration via a global entropic divergence loss, providing coarse-grained control over token-level probabilities.

3. We propose a novel non-verifiable reward that minimizes the hyper-volume among image, query, and response representations, and explore aggregation strategies beyond arithmetic summation, identifying harmonic aggregation as superior.

4. We evaluate our approach on the ViRFT benchmark, which includes tasks such as reasoning grounding, classification, and detection. We also implement additional RL-based baselines for comparison. Our proposed framework, DEVA, achieves significant improvements over these baselines (Fig. 3) and is supported by extensive ablation studies and analyses.

## 2 RELATED WORK

**Multimodal Large Language Models** (MLLMs) are an extension of large language models, where the model also processes visual input in addition to text input to produce a description of the visual input based on the query. One of the very popular MLLM includes GPT-4o (OpenAI, 2024), which produces excellent image understanding and reasoning. There are also other family of MLLMs that processes both images and text (Wang et al., 2024; Li et al., 2024; Zhang et al., 2024; Liu et al., 2024a). Recently, the MLLMs are involved in a two-stage training procedure: (a) pre-training and (b) post-training. The post-training stage is to specialize the MLLM to a particular task i.e. math, coding, perception. The post-training stage can involve fine-tuning using either supervised fine-tuning (SFT) or reinforcement learning (RL). RL has been useful to improve performance as well as instruction following and reasoning abilities. This has shown significant improvement in performance for LLMs Ziegler et al. (2019); Stiennon et al. (2022); Ouyang et al. (2022a); Ramamurthy et al. (2023); Zang et al. (2024); Carta et al. (2023); Sun et al. (2024); Snell et al. (2023); Abdulhai et al. (2023); Zhou et al. (2024a); Yao et al. (2023). Recently, RL especially GRPO and their variants have been used explicitly for multimodal perception and reasoning tasks Liu et al. (2025b); Zhou et al. (2025); Huang et al. (2025); Tan et al. (2025). In this paper, we mainly focus on improving GRPO and its variants for multimodal perception.

**Reinforcement Fine-tuning** Recently, there has been advent of reasoning models like Ope­nAI's o1 Jaech et al. (2024), which produced substantial improvement in performance in rea­soning. This has been enabled through rein­forcement learning (RL) techniques. In do­main of LLMs, there has been studies that have explored improving LLMs' performance in reasoning tasks such as solving mathemat­ical problems (Shao et al., 2024; Yang et al., 2024; Ying et al., 2024; Cai et al., 2024; Luong et al., 2024) and coding (Shao et al., 2024; Yang et al., 2024; Ying et al., 2024; Cai et al., 2024; Luong et al., 2024). Recently, there has been breakthrough in Deepseek-R1-Zero (Guo et al.,

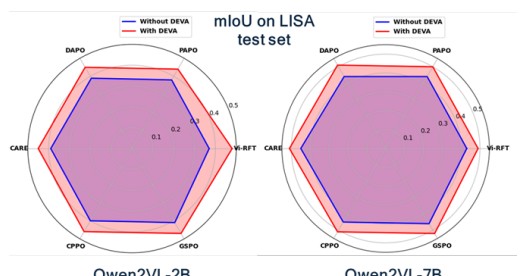

Figure 3: Effect of applying DEVA to various RL algorithms with Qwen2VL-2B & 7B model on LISA test set.

2025), which showed that using RL (GRPO) can be enough to produce reasoning capabilities. For MLLMs, RL has traditionally been used for reducing hallucinations and human preference model­ing (Sun et al., 2023a; Zhao et al., 2023; Zhou et al., 2024b; Sun et al., 2023b; Yu et al., 2024b; Liu et al., 2024d; Yu et al., 2024c; Zhou et al., 2024c). However, there were still some gaps on enhancing reasoning and visual perception of MLLMs. Recently, ViRFT (Liu et al., 2025b) was introduced that applied GRPO to a broad range of visual perception tasks. There has been a plethora of works ad­dressing the limitations of GRPO. These include PPO (Schulman et al., 2017b), PAPO (Wang et al., 2025b), DAPO (Yu et al., 2025), Dr GRPO (Liu et al., 2025a), BNPO (Xiao et al., 2025), GRPO-CARE (Chen et al., 2025), CPPO (Lin et al., 2025) and GSPO (Zheng et al., 2025). Our method also falls in this category where we improve reward diversity, aggregation and policy exploration. Our proposed framework is shown in 4.

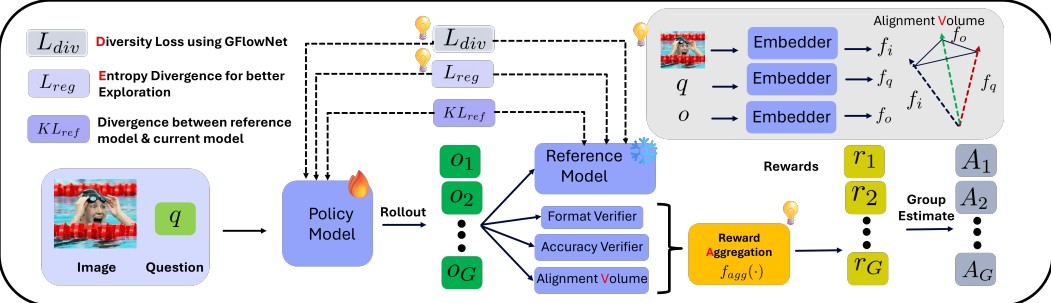

Figure 4: Our proposed DEVA framework can be applied on top of GRPO and its variants. Our contributions are highlighted with light bulb. This includes (a) diversity loss $L_{div}$ to improve reward diversity (b) regularization loss $L_{reg}$ to improve exploration ability of policy model. (c) Alignment volume for synchronized alignment of image, question and responses. (d) Reward aggregation.

## 3  METHODOLOGY

### 3.1  BACKGROUND

We introduce the Group Relative Policy Optimization (GRPO) algorithm (Shao et al., 2024), which differs from methods like PPO (Schulman et al., 2017a) that requires a separate critic model, making fine-tuning inefficient. GRPO avoids a critic by comparing rewards of candidate responses for feedback, enabling efficient training. For an input query $q$, the policy generates responses $\{o_1, o_2, \ldots, o_G\}$ with rewards $\{r_1, r_2, \ldots, r_G\}$. Rewards are normalized into advantages to optimize the per-token objective:

$$J(\theta) = E_{[q,\{o_i\}]} \frac{1}{G} \sum_{i=1}^{G} \frac{1}{|o_i|} \left\{ \min\left[ \frac{\pi_\theta}{\pi_{\theta_{old}}} A_i, \text{clip}\left( \frac{\pi_\theta}{\pi_{\theta_{old}}}, 1-\epsilon, 1+\epsilon \right) A_i \right] - \beta D_{KL}\left[ \pi_\theta \| \pi_{ref} \right] \right\}$$

$$(1)$$

where $A_i = \frac{r_i - \text{mean}(\{r_j\})}{\text{std}(\{r_j\})}$. $\pi_{\theta_{old}}$ is the policy in previous iteration while $\pi_{ref}$ is reference policy before fine-tuning starts. GRPO is often paired with verifiable rewards (Lambert et al., 2024; Guo et al., 2025; Team et al., 2025), using direct verification function instead of a learned reward model (Ouyang et al., 2022b; Liu et al., 2024b; Zang et al., 2025). This checks prediction-ground truth matches, effective for tasks with objective outcomes like math and coding. For visual tasks, rule-based rewards apply: classification uses accuracy (1 for correct, 0 otherwise), while detection uses IoU and bounding box confidence. Both tasks include format reward to enforce structured outputs (e.g., `<think>`, `<answer>`).

### 3.2  IMPROVED REWARD DIVERSITY

We motivated in Fig. 1 that GFlowNet based training objectives improve diversity of rewards. Here, we proceed to describe it. We follow the notation in (Kwon et al., 2024). GFlowNets are derived from token-wise Markov Decision Process (MDP) such that $\langle \mathcal{S}, \mathcal{A}, f \rangle$. The state space $\mathcal{S}$ consists of tokens generated so far. The action space $\mathcal{A}$ is the vocabulary, from where the next token is generated. The transition function $f$ is string concatenation, that facilitates the autoregressive process of MLLMs by appending at input. This state transitions go on until model produces end-of-sequence (EOS) token ($\top$). To summarize, the trajectory obtained from this auto-regressive generation is denoted as $o = o_n := o_{1:n}\top$, which encompasses the automated output response. The initial state is denoted as $s_0 := q$, which is the question. The terminal state is denoted as $s_f := q; o$.

Our model policy operates within this MDP, which is structured as a directed acyclic graph (DAG) and enriched with a positive semi-definite function $F$ called *flow*. We have three types of special states in the graph: (a) initial state $s_0$, which does not have any parent (b) terminal states $s_f$ with no children (c) intermediate states $s$, which can have both parent and children states. The reward is generally defined on terminal states.

Flow is defined on trajectories $\tau := (s_0 \to s_1 \to \cdots \to s_n) \in \mathcal{T}$, as $F : \mathcal{T} \to \mathbb{R}_{\geq 0}$. The state flow $F(s)$ is the sum of flows through all trajectories that pass through state $s$. On the other hand, the edge flow $F(s \to s')$ is the sum of flows through all trajectories that include transition from $s$ to $s'$. Since, all trajectories begin as $s_0$, the flow $F(s_0)$ serves as a normalization constant $Z$ used to define a probability distribution over the graph.

A flow is Markovian if there is a distribution $\pi(\cdot \mid s)$ over children of each non-terminal state $s$ such that trajectory probability is given as $\pi(\tau) = \prod_{t=1}^{n} \pi(s_t \mid s_{t-1}) = F(\tau)/Z$. The forward model policy i.e. the probability distribution $\pi(s_{t+1} \mid s_t)$ allows sampling of state trajectories i.e. output tokens. There is a backward policy $\pi_B(s_{t+1} \mid s_t)$. Both these policies can be expressed as flows such that $\pi(s_{t+1} \mid s_t) = F(s_t \to s_{t+1})/F(s_t)$ and $\pi_B(s_t \mid s_{t+1}) = F(s_t \to s_{t+1})/F(s_{t+1})$.

GFlowNets are trained using objectives that are derived from balance conditions. These balance conditions ensure that the network flow aligns with the graph's underlying dynamics. In our case, we empirically found detailed balance condition to be effective. For the training objective, we consider $F$ in terms of $\pi$, i.e. $F(s) = r(s)/\pi(s_f \mid s)$ with the condition that $r(s) := F(s \to s_f) = F(s)\pi(s_f \mid s)$ for terminating states. Since, the auto-regressive text generation is forward only, we consider the backward transition as redundant i.e. $\pi_B(s' \mid s) = 1$. For a transition from state $s$ to $s'$, the condition can be expressed as $F(s)\pi(s' \mid s) = F(s')\pi_B(s \mid s')$ . This constraint is expressed as a loss function in the log space as follows:

$$L_{\text{div}}(F, \pi, \pi_B) = \sum_{s \to s' \in \mathcal{A}} \left( \log \frac{F(s)\pi(s' \mid s)}{F(s')\pi_B(s \mid s')} \right)^2$$

The above expression can be rewritten in terms of reward and the policy with the following assumptions $\pi_B(\cdot) = 1$ and $F(s) = r(s)/\pi(s_f \mid s)$ as

$$L_{\text{div}}(\pi; r) = \sum_{t=1}^{n-1} \left( \log \frac{r(o_t \mid o_{1:t-1})\pi(\top \mid o_{1:t+1})}{r(o_{t+1} \mid o_{1:t})\pi(\top \mid o_{1:t})} + \log \pi(o_{t+1} \mid o_t) \right)^2 \tag{2}$$

Here, $\pi(\cdot) = \pi_\theta(\cdot)$. The reward $r(o_t \mid o_{1:t-1})$ is defined from reference model policy $\pi_{\text{ref}}(\cdot)$ as

$$\log r(o_t \mid o_{1:t-1}) = \log \pi_{\text{ref}}(o_t \mid o_{1:t-1}) + \exp \left( \frac{1}{\gamma} \log \pi_{\text{ref}}(\top \mid o_{1:t-1}) \right) \tag{3}$$

This design of reward function is made such that model does not deviate that much from reference. The presence of $\top$ makes sure that model can terminate appropriately. The hyper-parameter $\gamma \in (0, 1]$ is used to control strength of reward signal.

## 3.3 BETTER EXPLORATION

As presented in Eq. 1, the GRPO loss function incorporates a KL divergence term between the current policy distribution and a reference policy distribution. This regularization operates at the token level, aiming to prevent excessive deviation from the reference distribution. However, because the loss is computed locally on a per-token basis, it inherently limits exploration of the overall policy space. To enable broader policy exploration, we introduce a sequence-level metric in place of the token-level metric. This shift ensures that divergence is assessed globally, allowing greater flexibility in token-level distributions. By default, we define the regularization term $L_{reg}$ as the mean squared error between the average entropy of the policy model and that of the reference model, expressed as

$$L_{\text{reg}} = \|\frac{1}{m} \sum_{t=1}^{m} H_t^\theta - \frac{1}{n} \sum_{t=1}^{n} H_t^{ref}\|_2^2 \tag{4}$$

Here, $L_{\text{reg}}$ is computed for each group element, where $m$ and $n$ denote the sequence lengths of the policy and reference models, respectively. $H_t^\theta$ and $H_t^{ref}$ represent the entropy of the $t^{th}$ token in the policy and reference outputs. Alternative regularization objectives have been considered in Appendix J.

## 3.4 ALIGNMENT HYPER-VOLUME AND REWARD AGGREGATION

We present the methodology for computing the alignment hyper-volume and its integration into the reward framework. To estimate the volume in a high-dimensional space, we embed the image ($i$), query ($q$), and response ($o$) into a shared representation space. Notably, full image information is unnecessary; instead, a mask ($m$) is applied to extract relevant patches. This mask is derived by thresholding self-attention scores obtained from image–text token interactions within the language decoder (details in Appendix G). The embeddings are then computed using an encoder (e.g., a foundational model) as follows:

$$f_i = \Phi_i(i'), \quad f_q = \Phi_q(q), \quad f_o = \Phi_o(o), \quad \text{where} \quad i' = i \circ m \tag{5}$$

where $\circ$ denotes the Hadamard product. The resulting representations $f_i$, $f_q$, and $f_o$ are normalized to unit norm, constraining them to the surface of a unit hypersphere. The enclosed volume of the induced parallelotope is then computed via the determinant of the Gram matrix $G$:

$$V = \text{Vol}(f_i, f_q, f_o) = (\det G(f_i, f_q, f_o))^{1/2}, \quad G(f_i, f_q, f_o) = \begin{bmatrix} f_i \cdot f_i & f_i \cdot f_q & f_i \cdot f_o \\ f_q \cdot f_i & f_q \cdot f_q & f_q \cdot f_o \\ f_o \cdot f_i & f_o \cdot f_q & f_o \cdot f_o \end{bmatrix}. \tag{6}$$

Here, $\cdot$ denotes the inner product and $\det$ the determinant. The objective is to minimize $V$. To convert this into a reward $r_v$, we adopt an inverse relationship: $r_v = \max((aV^{-1} - b)^2, c)$, where $a$, $b$, and $c$ are hyperparameters. Although $r_v$ is non-verifiable (lacking ground truth), it can be combined with verifiable rewards such as format reward $r_{form}$ and task reward $r_{task}$ via an aggregation function $f_{agg}$:

$$r = f_{agg}(r_{form}, r_{task}, r_v) \tag{7}$$

By default, $f_{agg}$ is the arithmetic sum, though we also explore alternatives such as scaled geometric mean, scaled harmonic mean, and learned aggregation networks (pre-trained separately from the policy). Further details are provided in Appendix H.

## 4 EXPERIMENTS

### 4.1 EXPERIMENTAL SETUP

We follow the experimental setup and evaluation protocol of (Liu et al., 2025b) to examine whether our framework can adapt to and improve existing RL algorithms. We first consider few-shot learning, where the model is fine-tuned on a small number of samples for classification and detection tasks. Each sample consists of a triplet: question ($q$), image ($i$), and response ($o$). We compare against several RL baselines, including PPO (Schulman et al., 2017b), PAPO (Wang et al., 2025b), DAPO (Yu et al., 2025), Dr GRPO (Liu et al., 2025a), BNPO (Xiao et al., 2025), GRPO-CARE (Chen et al., 2025), CPPO (Lin et al., 2025), GMPO (Zhao et al., 2025), and GSPO (Zheng et al., 2025), along with a Chain-of-Thought (CoT) variant of supervised fine-tuning (SFT). Beyond few-shot tasks, we evaluate on the LISA dataset (Lai et al., 2024), which focuses on reasoning-based grounding, requiring the MLLM to interpret a query and predict bounding boxes for target objects. Following (Liu et al., 2025b), we use Qwen2-VL-2/7B (Wang et al., 2024). Additional results on few-shot and open-vocabulary detection using LVIS (Gupta et al., 2019), along with hyperparameter details, are provided in Appendix B.

### 4.2 FEW-SHOT CLASSIFICATION

We evaluate our approach on four fine-grained image classification datasets: Flower102 (Nilsback & Zisserman, 2008), Pets37 (Parkhi et al., 2012), FGVC-Aircraft (Maji et al., 2013), and Car196 (Krause et al., 2013). These datasets pose significant challenges due to the high visual similarity among categories. Table 1 reports the average accuracy across different shot settings for these datasets (values in parentheses), alongside COCO few-shot detection results.

As shown in Table 1, recent RL algorithms such as PAPO, GSPO, and GRPO-CARE outperform Visual-RFT, primarily due to sequence-level policy optimization strategies that optimize entire responses rather than token-wise references. PAPO, further specialize in multimodal settings by applying rewards to images and their perturbations. However, when DEVA and its ablations are applied on

Table 1: **Few-Shot results** We conducted 1-shot, 2-shot, 4-shot, 8-shot, and 16-shot experiments on 8 categories from COCO dataset. In paranthesis, we show the results for fine-grained classification dataset. Metric for COCO and fine-grained classification are mAP and accuracy respectively. We also report 4 shot results on COCO using Qwen2-VL 7B. Best is bold and second best is underlined.

| Model | 1-shot | 2-shot | 4-shot | 8-shot | 16-shot | 4-shot |
|---|---|---|---|---|---|---|
| Qwen2-VL-2B \| 7B (Wang et al., 2024) | 19.6 (56.0) | 19.6 (56.0) | 19.6 (56.0) | 19.6 (56.0) | 19.6 (56.0) | 43.0 |
| + SFT | 19.5 (51.7) | 21.0 (58.8) | 25.2 (55.6) | 30.2 (60.3) | 31.3 (64.0) | 44.1 |
| + SFT-CoT | 25.2 (59.2) | 27.7 (64.2) | 29.7 (66.4) | 34.7 (70.2) | 36.1 (74.2) | 48.2 |
| + PPO (Schulman et al., 2017b) | 31.2 (78.5) | 38.5 (78.4) | 36.7 (79.2) | 40.3 (81.4) | 43.9 (81.6) | 51.6 |
| + PAPO (Wang et al., 2025b) | 34.0 (81.1) | 42.0 (84.2) | 41.2 (81.9) | 48.0 (85.9) | 47.2 (86.2) | 55.1 |
| + DAPO (Yu et al., 2025) | 33.9 (81.3) | 41.8 (83.9) | 41.0 (82.3) | 47.7 (86.2) | 46.9 (86.6) | 55.0 |
| + Dr GRPO (Liu et al., 2025a) | 34.3 (82.2) | 42.3 (84.5) | 41.5 (83.0) | 47.9 (86.5) | 47.8 (86.6) | 55.6 |
| + BNPO (Xiao et al., 2025) | 34.2 (82.0) | 42.3 (85.1) | 40.8 (82.9) | 46.7 (87.0) | 47.5 (87.4) | 54.2 |
| + GRPO-CARE (Chen et al., 2025) | 34.7 (82.5) | 43.0 (85.5) | 41.7 (83.5) | 47.5 (86.7) | 48.3 (87.1) | 55.3 |
| + CPPO (Lin et al., 2025) | 34.3 (81.9) | 43.1 (86.7) | 42.7 (83.8) | 47.9 (87.3) | 48.2 (86.9) | 55.9 |
| + GMPO (Zhao et al., 2025) | 34.3 (81.5) | 42.4 (84.0) | 41.2 (83.0) | 47.2 (85.9) | 47.1 (86.5) | 54.5 |
| + GSPO (Zheng et al., 2025) | 35.0 (82.6) | 43.3 (85.2) | 42.6 (84.0) | 48.9 (87.8) | 48.3 (88.0) | 56.0 |
| + Visual-RFT (Liu et al., 2025b) | 33.6 (80.3) | 41.5 (83.5) | 40.6 (81.9) | 47.4 (85.1) | 46.8 (85.3) | 54.3 |
| + **DEVA** (**D**iv.) | 36.8 (83.0) | 44.2 (86.9) | 43.9 (84.5) | 49.9 (88.2) | 49.5 (88.7) | 57.6 |
| + **DEVA** (**D**iv. + **E**xplor.) | 37.9 (84.1) | 45.6 (87.2) | 45.0 (85.1) | 50.8 (88.8) | 50.1 (89.0) | 58.2 |
| + **DEVA** (**D**iv. + **E**xplor. + **A**lign. **V**ol.) | 38.9 (85.2) | 46.8 (88.0) | 46.2 (86.3) | 51.7 (89.4) | 51.2 (89.8) | 59.1 |
| + **DEVA** (**D**iv. + **E**xplor. + **A**lign. **V**ol. + **A**gg.) | 40.0 (86.1) | 47.9 (88.8) | 47.3 (87.1) | 52.9 (90.0) | 52.8 (91.1) | 60.0 |

top of Visual-RFT (vanilla GRPO), they consistently outperform all baselines. Even incorporating only diversity loss surpasses the strong GSPO baseline. Our full framework—combining diversity loss, exploration regularization, alignment volume reward, and harmonic aggregation—achieves a substantial **5–6 point** improvement over Visual-RFT and **3–4 points** over GSPO. Notably, gains from DEVA persist across increasing shot sizes. Additionally, incorporating CoT-style reasoning during fine-tuning further enhances performance compared to standard SFT.

## 4.3 FEW-SHOT OBJECT DETECTION

We extend evaluation to few-shot object detection. Specifically, we select eight COCO classes and vary number of fine-tuning samples per class (1, 2, 4, 8, and 16) to construct highly data-constrained training sets. Qwen2-VL-2B is fine-tuned across all settings, while Qwen2-VL-7B is fine-tuned for 4-shot case. The mean Average Precision (mAP) across all categories is reported in Table 1.

Both SFT and RL-based fine-tuning methods consistently outperform the baseline Qwen2-VL-2B and Qwen2-VL-7B models, following trends similar to those observed in few-shot classification. Our CoT-based dataset curation strategy yields an additional **4–6 point** improvement over vanilla SFT. While RL-based methods surpass Visual-RFT, our proposed DEVA framework, when applied on top of Visual-RFT, delivers substantial gains. Specifically, the full DEVA configuration achieves a **5–6 point** improvement over Visual-RFT, and in the 4-shot Qwen2-VL-7B setting, DEVA provides approximately **+6 points** over Visual-RFT and outperforms GSPO by about **+4 points**. Importantly, each component of DEVA—diversity loss, exploration regularization, alignment volume reward, and aggregation—contributes meaningfully, with each addition yielding roughly **+1 point** improvement.

## 4.4 REASONING GROUNDING

Here, we consider task of reasoning grounding, where the goal is to ground an object of interest depending on query. This kind of task is generally difficult for specialized models which cannot process and understand user's question. To address this task, the LISA (Lai et al., 2024) benchmark was introduced. We finetune both Qwen2-VL 2B/7B model (Wang et al., 2024) on small-scale dataset of 239 samples. On this setup, we evaluate specialized models, SFT methods as well as RL methods. The results of comparison studies are shown in Table 2.

From results, we see that the zero-shot quantitative performance of Qwen2-VL-2B and Qwen2-VL-7B, have improved performance compared to specialized models like OV-Seg, X-Decoder and GroundedSAM. Furthermore, SFT-CoT produces improved performance compared to SFT. As expected, our DEVA framework when applied on top of Visual-RFT produces staggering **+5-13pts** improvement. Furthermore, we see that improvement from just using diversity loss itself produces

Table 2: **Reasoning Grounding Results on LISA Lai et al. (2024)**. Visual-RFT surpasses SFT in reasoning grounding with 239 training images. We show results for both Qwen2-VL-2B and 7B. Best is bold and second best is underlined.

| Model | mIoU$_{test}$ | mIoU$_{val}$ | gIoU$_{test}$ | mIoU$_{test}$ | mIoU$_{val}$ | gIoU$_{test}$ |
|---|---|---|---|---|---|---|
| OV-Seg (Liang et al., 2023) | 28.4 | 30.5 | 26.1 | 28.4 | 30.5 | 26.1 |
| X-Decoder (Zou et al., 2023) | 28.5 | 29.1 | 24.3 | 28.5 | 29.1 | 24.3 |
| GroundedSAM (Liu et al., 2024c) | 26.2 | 28.6 | 21.3 | 26.2 | 28.6 | 21.3 |
| Qwen2-VL-2B $\mid$ 7B (Wang et al., 2024) | 26.9 | 30.1 | 25.3 | 40.4 | 45.2 | 38.0 |
| + SFT | 28.3 | 29.7 | 25.3 | 39.1 | 43.9 | 37.2 |
| + SFT-CoT | 30.3 | 33.7 | 28.3 | 40.5 | 45.4 | 38.9 |
| + PPO (Schulman et al., 2017b) | 33.6 | 36.9 | 33.2 | 41.3 | 46.1 | 40.1 |
| + PAPO (Wang et al., 2025b) | 38.2 | 41.4 | 35.6 | 44.2 | 47.9 | 43.8 |
| + DAPO (Yu et al., 2025) | 39.4 | 42.6 | 37.2 | 44.7 | 48.1 | 43.7 |
| + Dr GRPO (Liu et al., 2025a) | 38.2 | 41.4 | 36.3 | 44.3 | 48.5 | 44.0 |
| + BNPO (Xiao et al., 2025) | 38.1 | 41.3 | 37.0 | 44.5 | 48.9 | 44.2 |
| + GRPO-CARE (Chen et al., 2025) | 39.4 | 42.6 | 36.1 | 45.1 | 49.1 | 44.7 |
| + CPPO (Lin et al., 2025) | 40.1 | 43.3 | 36.2 | 45.6 | 49.2 | 45.0 |
| + GMPO (Zhao et al., 2025) | 40.5 | 43.2 | 35.6 | 44.7 | 46.2 | 43.5 |
| + GSPO (Zheng et al., 2025) | 41.3 | 44.5 | 37.1 | 46.0 | 49.9 | 46.1 |
| + Visual-RFT (Liu et al., 2025b) | 37.6 | 34.4 | 34.4 | 43.9 | 47.1 | 43.7 |
| + DEVA (**D**iv.) | 43.6 | 44.8 | 39.4 | 46.1 | 50.1 | 47.0 |
| + DEVA (**D**iv. + **E**xplor.) | 44.7 | 45.9 | 40.1 | 47.2 | 51.6 | 47.8 |
| + DEVA (**D**iv. + **E**xplor. + Align. **V**ol.) | _46.7_ | _46.9_ | _41.3_ | _48.1_ | _52.8_ | _48.2_ |
| + DEVA (**D**iv. + **E**xplor. + Align. **V**ol. + **A**gg.) | **48.9** | **47.3** | **42.3** | **49.5** | **53.5** | **48.9** |

higher improvement in performance compared to the highly competitive GSPO. The trend is repeated for the Qwen2-VL-7B model as well.

## 4.5 ADDITIONAL ANALYSES

We show visualization results on fine-grained classification and reasoning grounding in Fig. 5. From results, we see both SFT and Visual-RFT fail to identify class or localize object. For classification, both GSPO and DEVA (applied on Visual-RFT) produces correct responses with reasonable reasoning traces. For visual grounding, GSPO produces correct reasoning traces but still produces incorrect localization. DEVA produces correct reasoning trace as well as more compact bounding boxes.

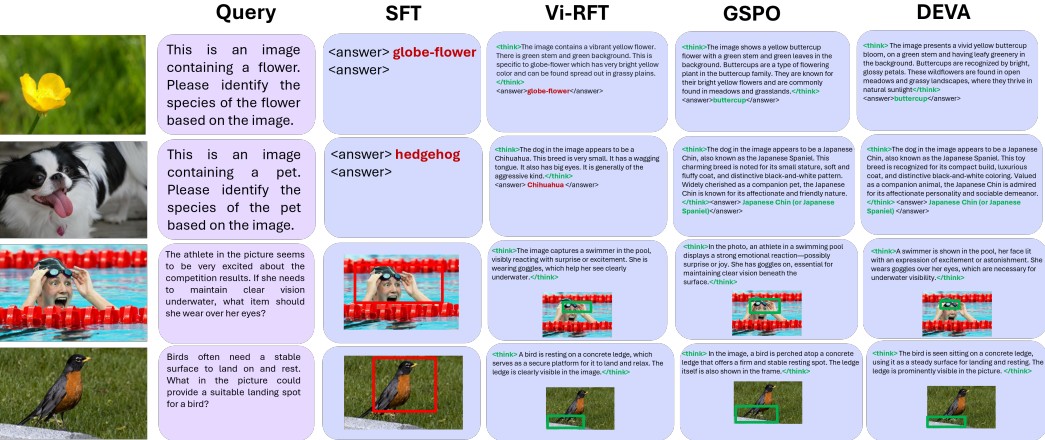

Figure 5: Top two rows are for classification task and bottom two rows are for reasoning grounding. Comparison is done across SFT, Visual-RFT, GSPO and DEVA. Results on SFT and Vi-RFT are reproduced.

In Figure 6 (a), we show attention visualization using the tools introduced in (Zhang et al., 2025). The fine-grained classification task is shown on the first two rows. The goal is to identify the model of the plane and car respectively. From the attention plot, we can see that the heatmaps generated from SFT, Visual-RFT and GSPO focus a lot on the background. On the other hand, the heatmap for DEVA focuses on the interior and the edges to better identify the model of the plane or the car. For the visual grounding task as well, the heatmap focuses more on the object of interest. For example,

in the third row DEVA can compactly localize the hammock. Similarly, DEVA can better localize the truck in the background instead of the car which is being incorrectly localized by other methods.

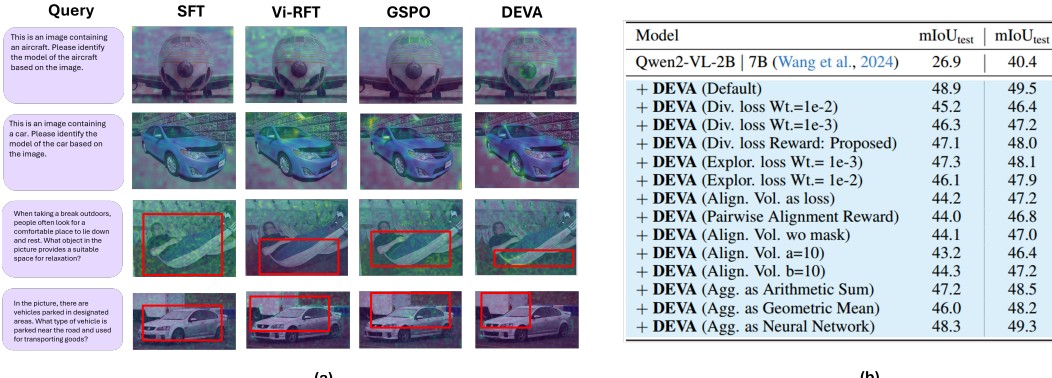

| Model | mIoU$_{test}$ | mIoU$_{test}$ |
|---|---|---|
| Qwen2-VL-2B \| 7B (Wang et al., 2024) | 26.9 | 40.4 |
| + **DEVA** (Default) | 48.9 | 49.5 |
| + **DEVA** (Div. loss Wt.=1e-2) | 45.2 | 46.4 |
| + **DEVA** (Div. loss Wt.=1e-3) | 46.3 | 47.2 |
| + **DEVA** (Div. loss Reward: Proposed) | 47.1 | 48.0 |
| + **DEVA** (Explor. loss Wt.= 1e-3) | 47.3 | 48.1 |
| + **DEVA** (Explor. loss Wt.= 1e-2) | 46.1 | 47.9 |
| + **DEVA** (Align. Vol. as loss) | 44.2 | 47.2 |
| + **DEVA** (Pairwise Alignment Reward) | 44.0 | 46.8 |
| + **DEVA** (Align. Vol. wo mask) | 44.1 | 47.0 |
| + **DEVA** (Align. Vol. a=10) | 43.2 | 46.4 |
| + **DEVA** (Align. Vol. b=10) | 44.3 | 47.2 |
| + **DEVA** (Agg. as Arithmetic Sum) | 47.2 | 48.5 |
| + **DEVA** (Agg. as Geometric Mean) | 46.0 | 48.2 |
| + **DEVA** (Agg. as Neural Network) | 48.3 | 49.3 |

(a)             (b)

Figure 6: This figure consists of (a) Attention visualization for the classification and reasoning grounding (b) Quantitative performance of different variants on reasoning grounding.

In Fig. 6 (b), we consider different variations and design choices of our DEVA framework. We see that on alternative design choices, the performance always drop on the test set when using both Qwen2-VL-2B and Qwen2-VL-7B. Also, we see that drop in performance when using a learnable aggregation method (i.e. a neural network) is minimal compared to that when using default harmonic aggregation, hinting that performance of the heuristic aggregation strategy matches that of the learned aggregation strategy. Furthermore, we see larger drop in performance when using alternative alignment schemes especially, when using pairwise alignment reward or using different hyper-parameters for the alignment volume reward.

In Fig. 7, we observe how the mIoU varies for different epochs for different RL algorithms as different components of the DEVA framework are added to the RL algorithm. The results are reported on the test set of the LISA dataset. Overall, we see that all the components of the DEVA framework are important and they lead to improved performance across all checkpoints.

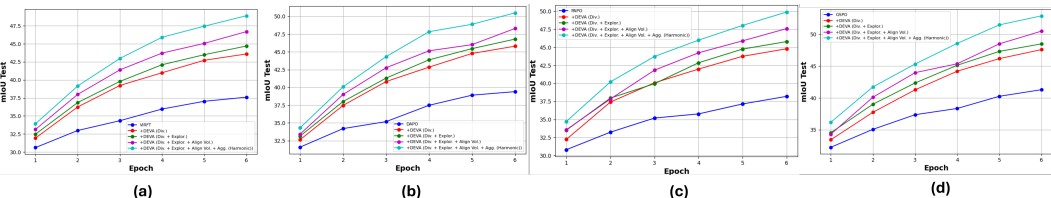

(a)          (b)          (c)          (d)

Figure 7: Performance (LISA dataset) over different epochs for different RL algorithms.

## 5 CONCLUSION

In this work, we presented DEVA, a novel framework designed to enhance GRPO and its variants when employed as training objectives for adapting MLLMs to visual perception tasks. We focus on challenging tasks such as fine-grained classification, object detection, and reasoning-based grounding. DEVA integrates four key components: (a) a diversity loss to enrich reward diversity and strengthen training signals, (b) an entropy-based divergence loss as a principled alternative to KL divergence for improved policy exploration, (c) an alignment volume reward to better align MLLMs with perception objectives, and (d) an optimal reward aggregation strategy for robust reward modeling. Our analysis reveals that each component plays a critical role in boosting Visual-RFT performance across tasks, both quantitatively and qualitatively. Notably, even introducing diversity loss alone surpasses the strong GSPO baseline. Furthermore, attention map visualizations demonstrate that DEVA achieves superior object localization compared to existing methods. Looking ahead, we aim to extend DEVA to more complex visual-agentic tasks.

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

## A    USE OF LARGE LANGUAGE MODEL

In this paper, we use GPT V only for refining and polishing the text in the paper.

## B    IMPLEMENTATION DETAILS

For datasets and benchmarking, we follow exact protocol and input prompts as introduced in (Liu et al., 2025b) following guidelines [2]. A few-shot learning approach is considered for image classification and object detection task and for the rest we consider fine-tuning on a small-scale dataset. For the SFT-CoT dataset, we generate a CoT reasoning dataset using Qwen2.5-VL-32B-Instruct (Bai et al., 2025) with the input as the image and the prompt as the following:

```
Question:  <QUERY>
Answer:  <OUTPUT>
Generate reasoning:  Explain step by step how to find the answer
from the image.
```

Here, `<QUERY>` and `<OUTPUT>` is replaced by the corresponding query and output in the training dataset. The generated reasoning is the ground-truth answer that will be used to train the model using SFT.

For fine-grained image classification benchmark, we consider four datasets: Flower102 (Nilsback & Zisserman, 2008), Pets37 (Parkhi et al., 2012), FGVC-Aircraft (Maji et al., 2013) and Car196 (Krause et al., 2013). For evaluation, we consider 1-shot, 2-shot, 4-shot, 8-shot and 16-shot protocol with the Qwen2-VL-2B model. For GRPO and their variants, we always use 8 generations. For all the datasets, we train for 8 epochs except for Pets37, where we train for 24 epochs. For training, we use a batch size of 8 distributed across 8 GPUs. We use bf16 datatype during the fine-tuning and gradient accumulation steps of 2. We follow the training setup in the lines of the

---

[2]https://github.com/Liuziyu77/Visual-RFT/tree/main

description [3]. For different baselines used for comparison, we use their open-source implementation and report results for the best hyper-parameter configuration. For DEVA, we use the following hyper-parameters: $\gamma = 0.5$ (Equation 3), the default weight on diversity $L_{\text{div}}$ and regularization loss $L_{\text{reg}}$ are both $1e-4$. The default values of $a$, $b$ and $c$ are 1.0, 0.0 and 2.0, respectively. For computing alignment features in Eq. 5, we use BLIP-2 Li et al. (2023) as the feature extractor, where we use the output of the Q-Former as the alignment features. These are then used to compute the alignment volume.

We also apply a few-shot learning setup for the object detection task. Specifically, we selected 8 classes from the COCO dataset and vary the number of fine-tuning samples per class. This includes 1, 2, 4, 8, and 16 training samples per class. This is done to construct training sets with very limited data. For this setup, we finetune Qwen2-VL-2B, while we fine-tune the Qwen2-VL-7B for the 4-shot case. The mAP for the 8 classes is calculated and the average is reported. The eight classes taken from the COCO dataset includes: bus, train, fire hydrant, stop sign, cat, dog, bed, toilet. The hyper-parameters are the same as that of fine-grained classification task except that for the 7B model, we use 4 generations for computing GRPO instead of 8.

We also evaluate on the LISA grounding benchmark, where the task is to ground the relevant part of an image given a query and an image. For the LISA grounding dataset, we finetune both the Qwen2-VL-2B and the Qwen2-VL-7B on 239 training samples. After finetuning is done, the model is then evaluated on the test and validation split of the LISA grounding benchmark. The hyper-parameters are the same as fine-grained classification except that fine-tuning is done for 6 epochs and for the 7B model, we use 4 generations instead of 8 for computing GRPO.

For Table 3, we use the same hyper-parameters as fine-grained classification except that for evaluation on the COCO dataset, the model is fine-tuned for 2 epochs while for evaluation using the LVIS dataset, the model is fine-tuned for 4 epochs.

## C  Additional Comparison Studies

Table 3: **Object Detection Results** First six columns show open vocabulary results on COCO dataset. We trained on 65 base categories and tested on 15 novel categories. Seventh and ninth column show few-shot results on LVIS dataset of 6 rare categories. We conducted 10-shot experiments on 6 rare categories from the LVIS dataset. Eighth and tenth column shows open vocabulary object detection results on LVIS dataset. We trained on the 65 base categories of the COCO dataset and tested on the 13 rare categories of the LVIS dataset. The parenthesis in the last column of the first row are the results of GroudingDINO-B (Liu et al., 2024c). Best results are shown in bold and second best results are underlined.

| Models | $mAP_n$ | $mAP_b$ | $mAP_{all}$ | $mAP_n$ | $mAP_b$ | $mAP_{all}$ | $mAP$ | $mAP$ | $mAP$ | $mAP$ |
|---|---|---|---|---|---|---|---|---|---|---|
| Qwen2-VL-2B \| 7B \| \| 2B \| 7B | 9.8 | 6.0 | 6.7 | 26.3 | 17.5 | 19.2 | 4.0 | 2.7 | 15.4 | 15.7 (**23.9**) |
| + SFT | 13.6 | 7.8 | 8.9 | 25.7 | 17.5 | 19.0 | 10.0 | 7.6 | 27.6 | 24.0 |
| + SFT-CoT | 17.1 | 12.8 | 12.2 | 29.3 | 20.8 | 22.1 | 13.5 | 12.2 | 28.9 | 27.4 |
| + PPO (Schulman et al., 2017b) | 27.6 | 16.2 | 17.3 | 33.2 | 23.1 | 24.9 | 16.3 | 17.2 | 30.1 | 28.5 |
| + PAPO (Wang et al., 2025b) | 32.2 | 21.3 | 24.6 | 37.0 | 27.9 | 28.3 | 22.4 | 22.1 | 35.2 | 32.3 |
| + DAPO (Yu et al., 2025) | 32.3 | 21.6 | 25.7 | 36.2 | 27.8 | 27.1 | 23.1 | 22.0 | 35.4 | 32.0 |
| + Dr GRPO (Liu et al., 2025a) | 33.1 | 22.4 | 26.3 | 37.0 | 28.9 | 28.5 | 24.2 | 23.1 | 36.7 | 33.3 |
| + BNPO (Xiao et al., 2025) | 32.0 | 21.9 | 25.8 | 37.2 | 27.5 | 28.1 | 24.6 | 24.0 | 37.5 | 34.1 |
| + GRPO-CARE (Chen et al., 2025) | 33.6 | 23.1 | 27.2 | 38.0 | 28.9 | 29.5 | 25.6 | 24.2 | 36.6 | 33.0 |
| + CPPO (Lin et al., 2025) | 33.2 | 24.2 | 27.1 | 39.1 | 29.5 | 30.3 | 24.0 | 24.3 | 37.2 | 34.2 |
| + GMPO (Zhao et al., 2025) | 32.5 | 23.1 | 26.4 | 37.8 | 28.3 | 29.2 | 24.3 | 23.9 | 35.2 | 33.3 |
| + GSPO (Zheng et al., 2025) | 34.6 | 25.2 | 28.2 | 39.6 | 30.1 | 31.2 | 25.3 | 25.2 | 37.6 | 35.5 |
| + Visual-RFT (Liu et al., 2025b) | 31.3 | 20.6 | 22.6 | 35.8 | 25.4 | 27.4 | 19.4 | 20.7 | 33.8 | 30.4 |
| + **DEVA (Div.)** | 37.8 | 28.1 | 30.2 | 41.3 | 32.8 | 34.0 | 26.2 | 27.3 | 40.1 | 37.2 |
| + **DEVA (Div. + Explor.)** | 38.9 | 30.0 | 31.3 | 42.5 | 33.9 | 35.1 | 27.4 | 28.6 | 41.4 | 38.6 |
| + **DEVA (Div. + Explor. + Align. Vol.)** | 39.9 | 31.3 | 32.5 | 43.9 | 34.6 | 36.8 | 28.7 | 29.9 | 42.8 | 39.8 |
| + **DEVA (Div. + Explor. + Align. Vol. + Agg.)** | **41.9** | **32.3** | **33.3** | **45.0** | **35.7** | **38.1** | **30.1** | **32.0** | **43.9** | **41.2** |

In Table 3, we report additional results on the open-vocabulary setup. The goal of this setup is to understand whether reinforcement fine-tuning can aid in better generalization compared to supervised fine-tuning (SFT). Specifically, we finetune both Qwen2-VL-2B and Qwen2-VL-7B on 65 base categories and evaluate on 13 novel categories. We also evaluate on the base categories as well as combination of base and novel categories. As expected, we can see that when we apply DEVA on top of Visual-RFT, it produces an improvement of **+10-12 pts** improvement in mAP across novel categories, base categories and an aggregated set of categories. We can even outperform the highly competitive GSPO by **+ 5 pts** improvement in mAP.

---

[3] https://github.com/Liuziyu77/Visual-RFT/issues/97

Table 4: **Few-shot results on Fine-grained Classification dataset.** We evaluated four fine-grained image classification datasets when **DEVA** is added to existing RL algorithms.

| Model | 1-shot | 2-shot | 4-shot | 8-shot | 16-shot |
|---|---|---|---|---|---|
| Qwen2-VL-2B (Wang et al., 2024) | 56.0 | 56.0 | 56.0 | 56.0 | 56.0 |
| + PAPO (Wang et al., 2025b) | 81.1 | 84.2 | 81.9 | 85.9 | 86.2 |
| + PAPO + **DEVA** | 86.4 | 89.4 | 87.3 | 91.4 | 91.6 |
| + DAPO (Yu et al., 2025) | 81.3 | 83.9 | 82.3 | 86.2 | 86.6 |
| + DAPO + **DEVA** | 86.5 | 89.1 | 87.4 | 91.5 | 91.8 |
| + GRPO-CARE (Chen et al., 2025) | 82.5 | 85.5 | 83.5 | 86.7 | 87.1 |
| + GRPO-CARE + **DEVA** | 87.4 | 90.4 | 88.5 | 92.2 | 92.6 |
| + CPPO (Lin et al., 2025) | 81.9 | 86.7 | 83.8 | 87.3 | 86.9 |
| + CPPO + **DEVA** | 87.1 | 91.5 | 89.0 | 92.7 | 92.4 |
| + GSPO (Zheng et al., 2025) | 82.6 | 85.2 | 84.0 | 87.8 | 88.0 |
| + GSPO + **DEVA** | 87.8 | 90.6 | 89.3 | 93.2 | 93.4 |

Table 5: **Reasoning Grounding Results on LISA Lai et al. (2024)**. We evaluated reasoning grounding results when **DEVA** is added to existing RL algorithms.

| Model | mIoU$_{\text{test}}$ | mIoU$_{\text{val}}$ | gIoU$_{\text{test}}$ | mIoU$_{\text{test}}$ | mIoU$_{\text{val}}$ | gIoU$_{\text{test}}$ |
|---|---|---|---|---|---|---|
| Qwen2-VL-2B | 7B (Wang et al., 2024) | 26.9 | 30.1 | 25.3 | 40.4 | 45.2 | 38.0 |
| + PAPO (Wang et al., 2025b) | 38.2 | 41.4 | 35.6 | 44.2 | 47.9 | 43.8 |
| + PAPO + **DEVA** | 44.6 | 47.7 | 42.0 | 50.3 | 54.4 | 50.1 |
| + DAPO (Yu et al., 2025) | 39.4 | 42.6 | 37.2 | 44.7 | 48.1 | 43.7 |
| + DAPO + **DEVA** | 45.8 | 48.9 | 43.3 | 51.2 | 54.8 | 49.9 |
| + GRPO-CARE (Chen et al., 2025) | 39.4 | 42.6 | 36.1 | 45.1 | 49.1 | 44.7 |
| + GRPO-CARE + **DEVA** | 45.6 | 48.7 | 42.3 | 51.3 | 55.2 | 50.2 |
| + CPPO (Lin et al., 2025) | 40.1 | 43.3 | 36.2 | 45.6 | 49.2 | 45.0 |
| + CPPO + **DEVA** | 46.3 | 49.4 | 42.8 | 51.9 | 55.6 | 50.6 |
| + GSPO (Zheng et al., 2025) | 41.3 | 44.5 | 37.1 | 46.0 | 49.9 | 46.1 |
| + GSPO + **DEVA** | 47.2 | 50.3 | 43.4 | 52.3 | 56.2 | 51.4 |

We also evaluate the model trained on COCO on 13 rare categories of the LVIS dataset. This is shown in the eight and tenth column of the Table 3. DEVA essentially produces **+10 pts** improvement over Visual-RFT and **+5-6 pts** improvement over GSPO. Finally, we also evaluate 10-shot object detection performance within the LVIS dataset of 6 rare categories. We show similar improvement in performance compare to Visual-RFT and GSPO. To summarize, from Table 3, it is clear that DEVA is more effective for open vocabulary setup and can easily boost generalization capabilities.

## D  BOOSTING COMPETITIVE METHODS

In this section, we analyze whether DEVA can boost existing competitive methods. This is shown for fine-grained image classification dataset in Table 4 and for LISA reasoning grounding dataset in Table 5. In Table 4, we see that our proposed method can produce **+4-5 pts** improvement in accuracy when applied to existing RL algorithms. For reasoning grounding task in Table 4, we also observe similar trends, where our proposed framework can produce improvements upto **+6-7 pts** in IoU.

## E  EFFECT OF LOW RANK ADAPTATION

For adapting the multi-modal model on small-scale fine-tuning data, our default setup is to finetune the whole model. In this section, we consider the situation where we finetune LoRA instead of fine-tuning the whole model. We consider different variations for LoRA. This includes changing ranks for LoRA and also the attachment points. The adaptors are attached on the $Q, K, V$ matrices in the transformer layers of the large language model (LLM) and/or vision encoder (VE). The results are shown in Table 6. From the results, we see that fine-tuning LoRA instead of full finetuning

Table 6: **Reasoning Grounding Results on LISA (Lai et al., 2024)**. We evaluated reasoning rounding results when **DEVA** is added to existing RL algorithms and finetuned using LoRA (Hu et al., 2022).

| Model | mIoU$_{test}$ | mIoU$_{val}$ | gIoU$_{test}$ | mIoU$_{test}$ | mIoU$_{val}$ | gIoU$_{test}$ |
|---|---|---|---|---|---|---|
| Qwen2-VL-2B │ 7B (Wang et al., 2024) | 26.9 | 30.1 | 25.3 | 40.4 | 45.2 | 38.0 |
| + **DEVA** (Full Fine-tuning) | 48.9 | 47.3 | 42.3 | 49.5 | 53.5 | 48.9 |
| + **DEVA** (Rank = 16, Attach: LLM) | 45.2 | 44.0 | 38.2 | 46.2 | 50.3 | 45.3 |
| + **DEVA** (Rank = 32, Attach: LLM) | 45.7 | 44.5 | 38.8 | 46.9 | 50.9 | 46.0 |
| + **DEVA** (Rank = 64, Attach: LLM) | 46.1 | 45.2 | 39.5 | 47.6 | 51.5 | 46.9 |
| + **DEVA** (Rank = 16, Attach: VE) | 43.7 | 42.2 | 36.3 | 44.0 | 48.2 | 43.2 |
| + **DEVA** (Rank = 32, Attach: VE) | 44.2 | 43.0 | 37.0 | 44.8 | 48.9 | 43.9 |
| + **DEVA** (Rank = 64, Attach: VE) | 44.9 | 43.8 | 38.0 | 45.4 | 49.6 | 44.6 |
| + **DEVA** (Rank = 16, Attach: VE + LLM) | 46.5 | 45.9 | 40.5 | 48.0 | 52.0 | 47.1 |
| + **DEVA** (Rank = 32, Attach: VE + LLM) | 47.9 | 46.8 | 41.2 | 48.8 | 52.9 | 47.8 |
| + **DEVA** (Rank = 64, Attach: VE + LLM) | 48.5 | 47.1 | 41.9 | 49.1 | 53.4 | 48.5 |

produces subpar performance, which is expected since LoRA modifies a very small subspace of the parameter space compared to full finetuning. As expected, higher ranks for LoRA produces higher IoU since it closely approximates full fine-tuning. When LoRA is attached to both VE and LLM, visual perception capabilities for multimodal LLMs are enhanced better compared to attaching LoRA to either VE or LLM. Furthermore, results show that it is more effective to attach LoRA to LLM instead of VE. This might be because the LLMs are more responsible for multimodal reasoning tasks and need to be adapted to the specific visual grounding task. On the other hand, the VE is already capable in handling perception tasks. This empirical evidence has also been highlighted before in (Cocchi et al., 2025).

## F   EFFECT ON OTHER MODELS

In this subsection, we test whether our method is applicable to other models: GLM-Edge (GLM et al., 2024) and LLAVA (Liu et al., 2023) in Table 7. We observe that our framework DEVA produces significant improvement in IoU over Visual-RFT and also surpasses the IoU of GSPO. However, the gap between GSPO and our proposed DEVA framework is diminished for LLAVA1.5-7B. Overall, we see DEVA is more effective for smaller models. This suggests that our framework can be very effective for small-scale devices to be deployed on edge devices.

Table 7: **Reasoning Grounding Results on LISA Lai et al. (2024)**. using the GLM-Edge model (GLM et al., 2024). and LLAVA1.5 (Liu et al., 2023)

| Model | mIoU$_{test}$ | mIoU$_{val}$ | gIoU$_{test}$ | mIoU$_{test}$ | mIoU$_{val}$ | gIoU$_{test}$ |
|---|---|---|---|---|---|---|
| GLM-Edge-V-2B │ LLAVA1.5-7B | 24.4 | 27.5 | 22.5 | 38.9 | 42.3 | 35.4 |
| + SFT | 26.8 | 27.2 | 22.6 | 36.2 | 41.1 | 34.3 |
| + SFT-CoT | 28.6 | 31.2 | 25.6 | 38.3 | 42.7 | 36.2 |
| + PPO (Schulman et al., 2017b) | 31.1 | 34.4 | 30.4 | 39.2 | 43.4 | 37.3 |
| + PAPO (Wang et al., 2025b) | 35.4 | 38.7 | 32.9 | 42.0 | 45.3 | 41.0 |
| + DAPO (Yu et al., 2025) | 36.9 | 40.0 | 34.4 | 42.2 | 45.3 | 41.0 |
| + Dr GRPO (Liu et al., 2025a) | 35.4 | 38.7 | 33.5 | 41.5 | 45.7 | 41.2 |
| + BNPO (Xiao et al., 2025) | 35.3 | 38.6 | 34.2 | 41.7 | 46.1 | 41.4 |
| + GRPO-CARE (Chen et al., 2025) | 36.9 | 40.0 | 33.3 | 42.3 | 46.3 | 41.9 |
| + CPPO (Lin et al., 2025) | 37.6 | 40.8 | 33.4 | 42.8 | 46.4 | 42.2 |
| + GMPO (Zhao et al., 2025) | 36.3 | 40.5 | 32.9 | 41.8 | 46.2 | 41.8 |
| + GSPO (Zheng et al., 2025) | 38.5 | 42.0 | 34.3 | 43.2 | 47.1 | 43.3 |
| + Visual-RFT (Liu et al., 2025b) | 34.8 | 31.8 | 31.6 | 42.0 | 44.3 | 41.0 |
| + **DEVA** (**D**iv.) | 41.1 | 42.2 | 36.6 | 43.2 | 47.3 | 44.2 |
| + **DEVA** (**D**iv. + **E**xplor.) | 42.2 | 43.3 | 37.3 | 44.3 | 48.8 | 45.0 |
| + **DEVA** (**D**iv. + **E**xplor. + **A**lign. **Vol**.) | 44.2 | 44.4 | 38.5 | 45.2 | 50.0 | 45.4 |
| + **DEVA** (**D**iv. + **E**xplor. + **A**lign. **Vol**. + **Agg**.) | **46.4** | **44.8** | **39.5** | **46.6** | **50.7** | **46.1** |

## G  MASK COMPUTATION FOR MAPPING IMAGES FOR ALIGNMENT VOLUME

Our goal is to obtain the binary mask $m$ in the image for computing alignment reward defined in Eq. 5. For computing the binary mask $m$, we need to do a forward pass of the image and text query through the multi-modal large language model to obtain attention scores and backtrack them to the image to obtain relevant patches. The details of obtaining relevant patches are described below.

**Attention-to-patch mask.**   We consider that $i \in \mathbb{R}^{H \times W \times 3}$ is resized by the image processor to $i' \in \mathbb{R}^{H' \times W' \times 3}$. If we have patch size $p = 14$, the visual encoder yields a grid of patches of size $H_p = H'/p$, $W_p = W'/p$, and $N_v = H_p W_p$ visual tokens. In the multimodal sequence, visual tokens are arranged continuously as $\mathcal{V} = \{ tok \mid tok_{vs} < tok < tok_{ve} \}$ between special tokens at positions $tok_{vs}$ and $tok_{ve}$, respectively.

**Decoder attentions over visual tokens.**   At decoding step $t$ (when predicting text token $o_t$), layer $\ell \in \{1, \dots, L\}$ and head $h \in \{1, \dots, H\}$ produces self-attention matrix $A^{(\ell,h,t)} \in \mathbb{R}^{T_t \times T_t}$, whose row $t$ is distribution over source positions $i \in \{1, \dots, T_t\}$. We define an aggregated score such that

$$s_i^{(t)} = \sum_{\ell=1}^{L} \sum_{h=1}^{H} w_\ell\, u_h\, A_{t,i}^{(\ell,h,t)} \qquad (i \in \mathcal{V}), \tag{8}$$

with nonnegative weights $w_\ell, u_h$ such that $\sum_{\ell=1}^{L} w_\ell = 1$ and $\sum_{h=1}^{H} u_h = 1$.

We apply a min–max normalization over visual positions:

$$\tilde{s}_i^{(t)} = \frac{s_i^{(t)} - \min_{j \in \mathcal{V}} s_j^{(t)}}{\max_{j \in \mathcal{V}} s_j^{(t)} - \min_{j \in \mathcal{V}} s_j^{(t)} + \varepsilon} \in [0,1]. \tag{9}$$

**Aggregating multiple answer tokens.**   We consider the case when the mask considers multiple output tokens. In that case, we let $\mathcal{T}$ be the indices and $v_t \geq 0$ with $\sum_{t \in \mathcal{T}} v_t = 1$. We define

$$\tilde{s}_i = \sum_{t \in \mathcal{T}} v_t\, \tilde{s}_i^{(t)}. \tag{10}$$

**Mapping visual tokens to the patch grid.**   Index visual tokens locally as $k \in \{1, \dots, N_v\}$ (in order within $\mathcal{V}$). Map $k$ to patch coordinates $(r, c)$ via

$$r = 1 + \left\lfloor \frac{k-1}{W_p} \right\rfloor, \qquad c = 1 + \big((k-1) \bmod W_p\big). \tag{11}$$

Let $i(k)$ denote the absolute sequence index corresponding to the $k$-th visual token. The patch-level score map $S \in [0,1]^{H_p \times W_p}$ is

$$S_{r,c} = \tilde{s}_{i(k)}. \tag{12}$$

**Upsampling and binarization.**   Let $\mathcal{U}_p$ be bilinear upsampling by factor $p$. The soft mask over $I'$ is

$$M = \mathcal{U}_p(S) \in [0,1]^{H' \times W'}. \tag{13}$$

A binary mask at threshold $\tau \in (0,1)$ is given by

$$m(x,y) = \mathbf{1}[\, M(x,y) \geq \tau \,]. \tag{14}$$

The threshold $\tau$ is given as 0.5.

## H  DIFFERENT REWARD AGGREGATION TECHNIQUES

In this method, we consider different reward aggregation techniques like arithmetic mean, geometric mean, harmonic mean and neural network, etc. Let us consider that we have three types of rewards: format reward $r_{form}$, task reward $r_{task}$ and volume reward $r_v$. In that case, we consider the following types of aggregation.

**Arithmetic Sum:** For the scaled arithmetic mean, we consider the aggregated reward $r = (r_{form} + r_{task} + r_v)$.

**Geometric Mean:** For the scaled geometric mean, we consider the aggregated reward $r = 3(r_{form}r_{task}r_v)^{1/3}$

**Harmonic Mean:** For the scaled harmonic mean, we consider the aggregated reward $r = 9/((1/r_{form}) + (1/r_{task}) + (1/r_v))$

**Neural Network:** For the neural network $\Phi(\cdot)$, we consider the following formulation for prediction. It takes in the three reward scalars $r_{form}$, $r_{task}$ and $r_v$ and produces an aggregated reward $r$ such that $r = \Phi(r_{form}, r_{task}, r_v)$. When we use this neural network, it is a multi-stage training procedure:

- **Stage 1:** We train the policy model with the harmonic reward using the GRPO training objective for the same epoch numbers as standard fine-tuning.
- **Stage 2:** With the same GRPO training objective, we freeze the policy model and train the neural network based predictor that takes in three reward scalars to produce the desired reward. This training is done for half the number of epochs as standard fine-tuning.
- **Stage 3:** During the final stage, we freeze the neural network based predictor and fine-tune the policy model for the same number of epochs as standard fine-tuning.

The neural network architecture is two-layered with input size of 3, hidden state size of 2 and output size as 1.

## I   METRIC PROGRESSION CURVES

In this section, we report results on the LISA reasoning dataset, how different metrics progress over different iterations. It is important to note that the progression of rewards and their dynamics over training iterations does not always correlate proportionally with the final evaluation metric i.e. mIoU on visual grounding tasks.

In Figure 8, we observe how the total reward progresses over training iterations for different RL algorithms. The results are shown for different algorithms as the diversity loss ($L_{div}$) and exploration loss ($L_{reg}$) is added to existing loss term. We observe that as we add the diversity loss $L_{div}$, the reward curve (as shown by the orange curve) converges faster and reaches a higher saturation value. Similarly, as we add the exploration loss $L_{reg}$, the reward curve (as shown by the green curve) converges much faster and reaches a much higher value.

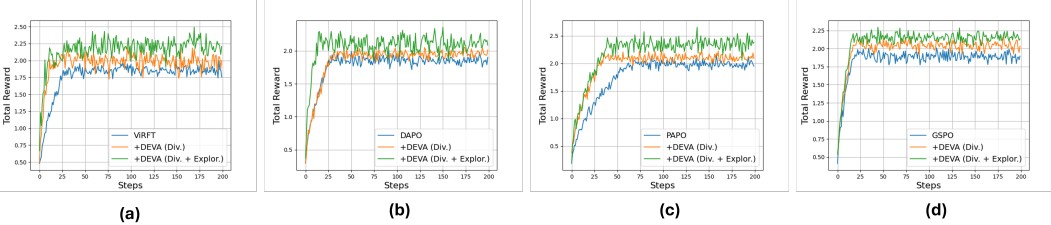

Figure 8: This figure shows how the total reward progresses for different RL algorithms as diversity loss $L_{div}$ and regularization loss $L_{reg}$ for improved exploration is added.

In Figure 9, we also observe how the total reward progresses over training iterations for different RL algorithms as our full framework DEVA is applied on top of existing RL algorithms like ViRFT, DAPO, PAPO and GSPO. We also try out different aggregation methods: (a) Arithmetic Sum (b) Scaled Geometric Mean (c) Scaled Harmonic Mean (d) Neural Network etc. From the results in Table 9 (a), we observe that the reward curve reaches different saturation values for different RL algorithms with GSPO producing the highest reward value. The difference in the reward curves for different methods diminishes when alternative aggregation methods are used. This suggests that there is consistency in the reward curves when different aggregation methods are used. Furthermore,

the rate of increase of the reward curve is the fastest when scaled harmonic mean and neural network is used.

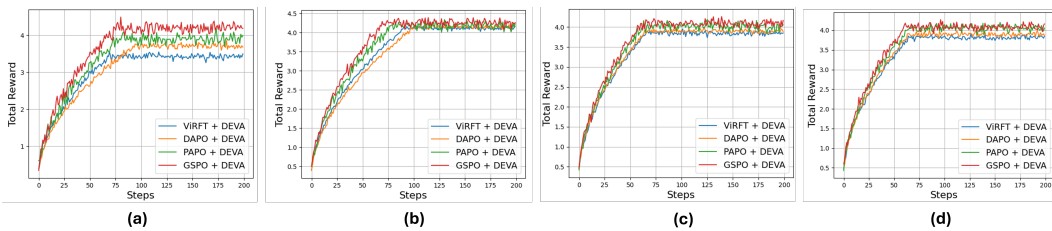

Figure 9: In this figure, we plot the reward curves when our full framework DEVA is applied to different RL algorithms. We consider different aggregation methods like (a) Arithmetic Sum (b) Scaled Geometric Mean (c) Scaled Harmonic Mean (d) Neural network

In Figure 10, we also observe how the KL divergence of token probabilities between the reference model and the current policy model varies with training steps for different methods. With the addition of diversity loss, there is a slight increase in KL divergence, when the diversity loss is added to each of the RL algorithms. This is because increasing the diversity produces more variable amount of reasoning traces and hence produces more higher range of KL divergence values. When the exploration loss is added to the RL algorithm, it further increases the KL divergence range leading to better exploration.

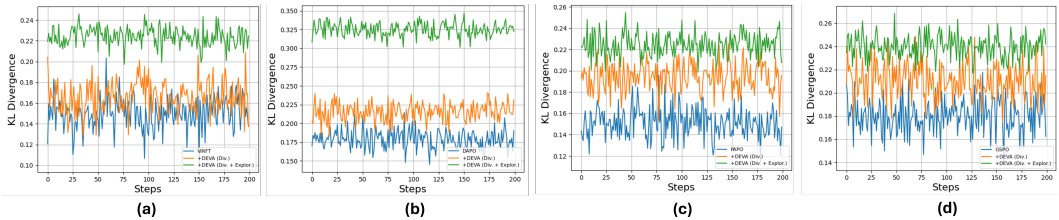

Figure 10: This figure shows how the KL divergence in Eq. 1 varies for different RL algorithms as diversity loss $L_{\text{div}}$ and regularization loss $L_{\text{reg}}$ for improved exploration is added.

## J ADDITIONAL HYPERPARAMETER STUDIES

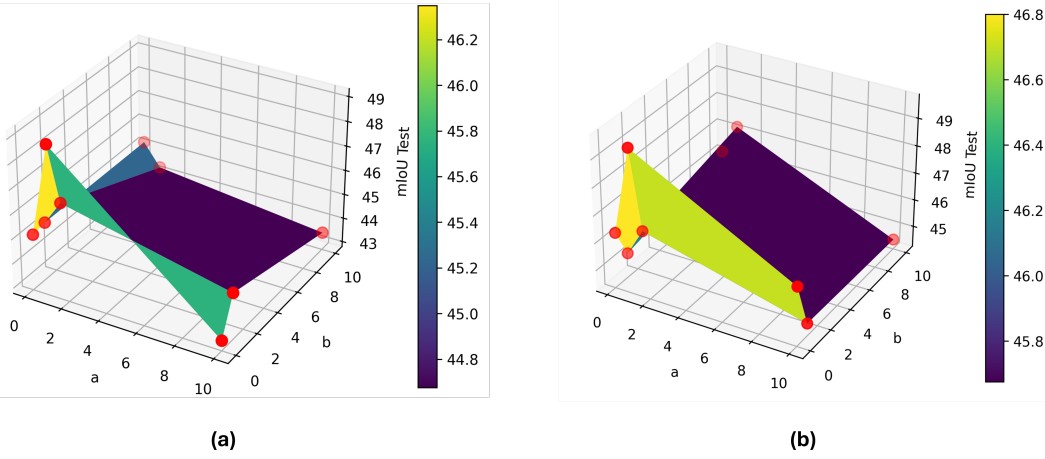

Figure 11: This figure consists of (a) Surface curve of mIoU for hyper-parameters $a$ and $b$ using Qwen-VL-2B on LISA reasoning dataset. (b) Surface curve of mIoU for hyper-parameters $a$ and $b$ using Qwen-VL-7B on LISA reasoning dataset.

In Fig. 11, we observe variation in mIoU on LISA test set for both (a) Qwen-VL-2B and (b) Qwen-VL-7B, when the hyper-parameters $a$ and $b$ are varied while $c$ is fixed at 2 for the volume reward alignment defined in Section 3.4 in the main section. As expected, the default setting of $a = 1.0$ and $b = 0.0$, produces the most optimal performance.

In Table 8, we observe how the mIoU varies for different variations. With respect to the entropy divergence loss defined in Eq. 4, we consider the following variations: (a) Partition 2: When mean squared error is computed separately for 2 partitions of the tokens of the reference model and policy model. (b) Partition 3: When mean squared error is computed separately for 3 partitions of the tokens of the reference model and the policy model. (c) OT: We consider the optimal transport distance (Courty et al., 2016) between the two entropy vectors obtained from the reference model and the policy model. The cost matrix is computed such that each element is the cosine distance between CLIP (Radford et al., 2021) embedding of the two words.

Furthermore, we consider feature extractors defined in Eq. 5. This includes CLIP (Radford et al., 2021) and SigLip2 (Tschannen et al., 2025). Overall, we observe that a model with larger capacity produces better performance. However, all model variants produce poorer performance compared to the default version of DEVA.

Table 8: **Reasoning Grounding Results on LISA Lai et al. (2024)**. Selected metrics are shown for different model variations.

| Model | $mIoU_{test}$ | $mIoU_{test}$ |
|---|---|---|
| Qwen2-VL-2B │ 7B (Wang et al., 2024) | 26.9 | 40.4 |
| + **DEVA** (Default) | 48.9 | 49.5 |
| + **DEVA** (Explor. Loss: Partition=2) | 48.0 | 48.7 |
| + **DEVA** (Explor. Loss: Partition=3) | 46.5 | 47.6 |
| + **DEVA** (Explor. Loss: OT) | 48.5 | 48.9 |
| + **DEVA** (Embed: CLIP B-16) | 46.3 | 47.1 |
| + **DEVA** (Embed: CLIP L-14) | 47.5 | 48.9 |
| + **DEVA** (Embed: SigLip2 B-16) | 46.9 | 47.8 |
| + **DEVA** (Embed: SigLip2 L-16) | 47.8 | 49.0 |