# K  ADDITIONAL VISUALIZATION

In this section, we show additional visualization for some samples on the fine-grained classification and LISA reasoning grounding datasets. We visualize the heatmaps as well as the bounding boxes for the LISA grounding datasets. From the results, we see that DEVA produces better localization capabilities compared to other methods, where the heatmap focuses more around the objects of interest. Furthermore, we observe that with DEVA, introducing additional components of the framework progressively produces better heatmap localization and focus.

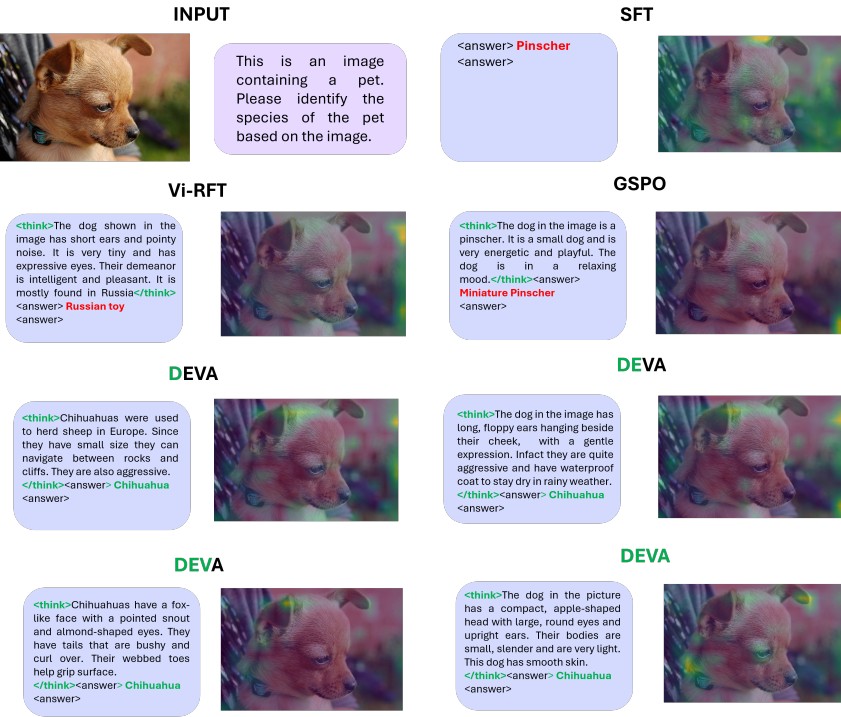

Figure 12:  Visualization of the Chihuahua as part of the Pets37 dataset.

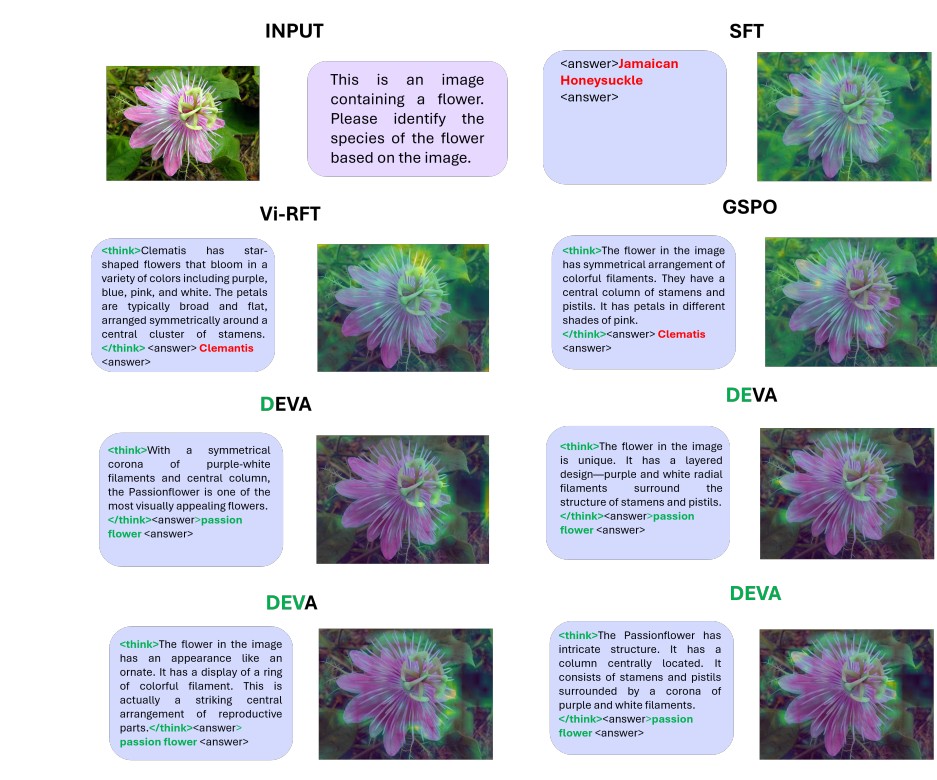

Figure 13: Visualization of the Passion Flower as part of the Flower102 dataset.

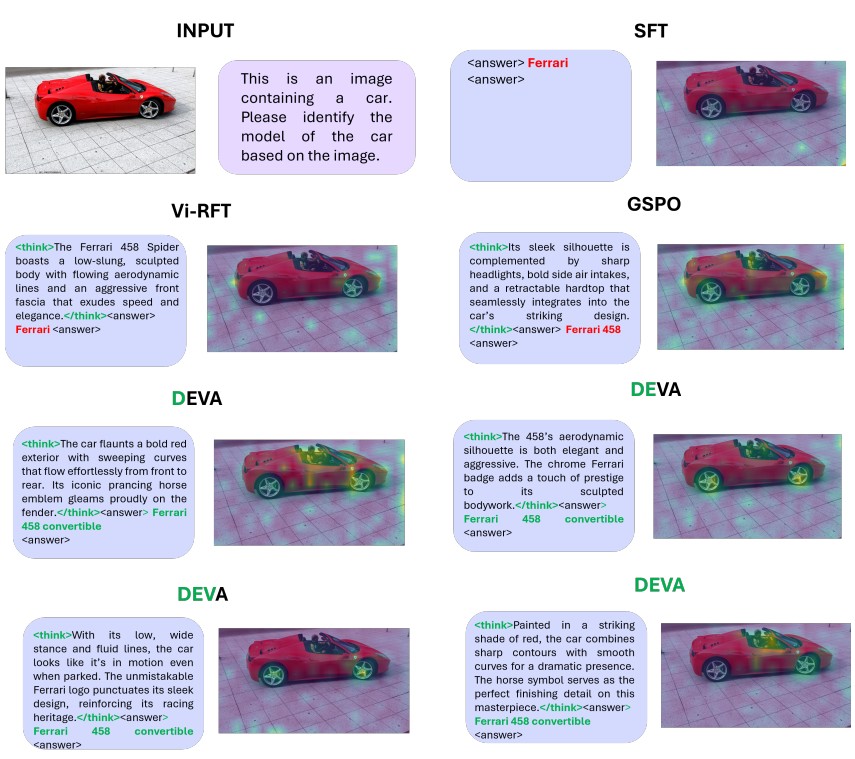

Figure 14: Visualization of the Ferrari car as part of the Stanford Cars dataset.

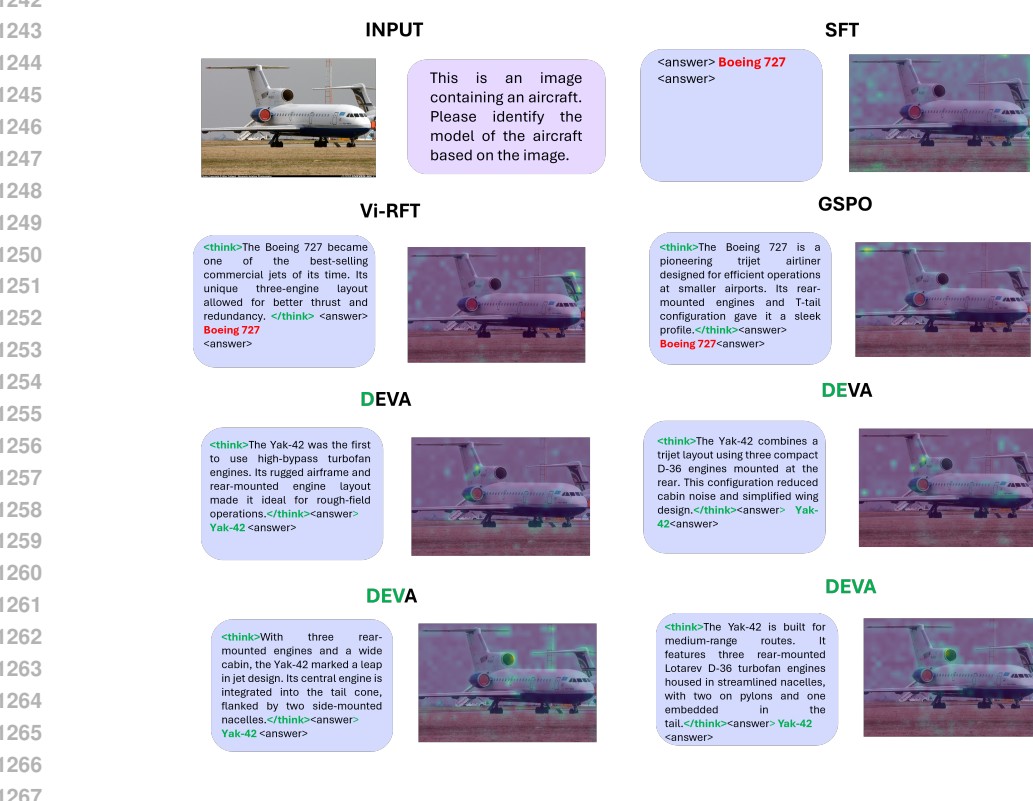

Figure 15: Visualization of the Airplane as part of the FGVC dataset.

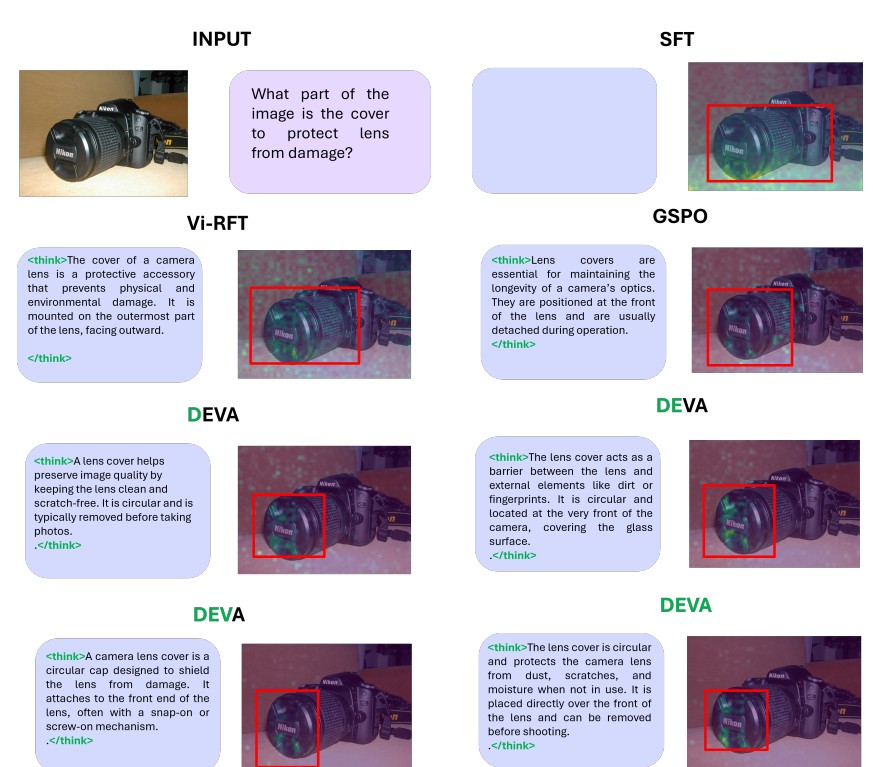

Figure 16: Visualization of the Camera as part of the LISA reasoning grounding dataset.

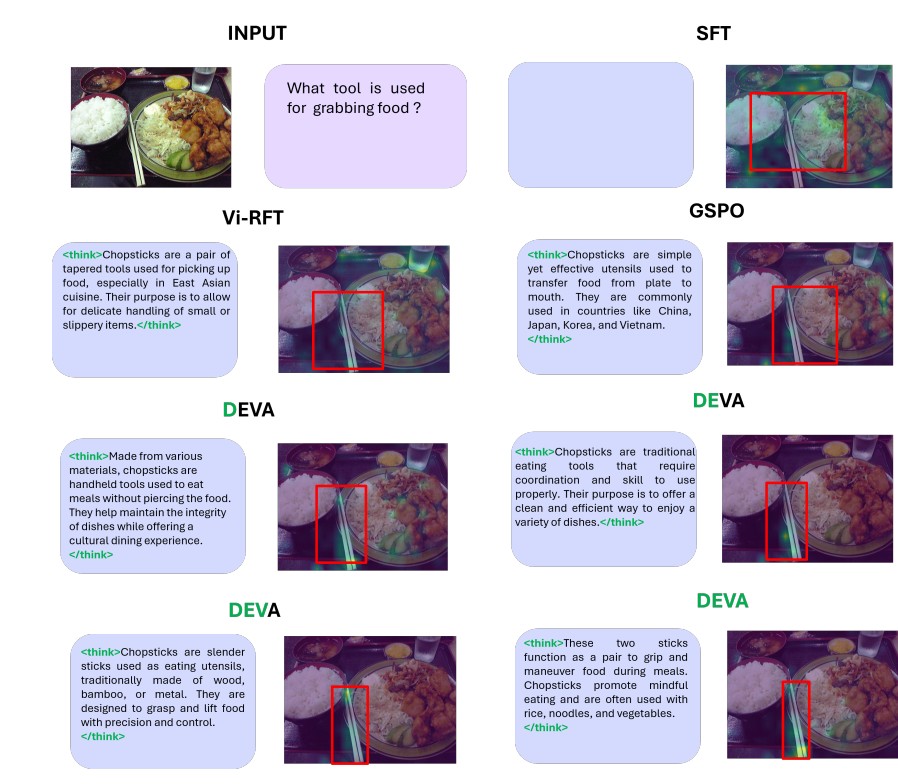

Figure 17: Visualization of the Chopstick as part of the LISA reasoning grounding dataset.

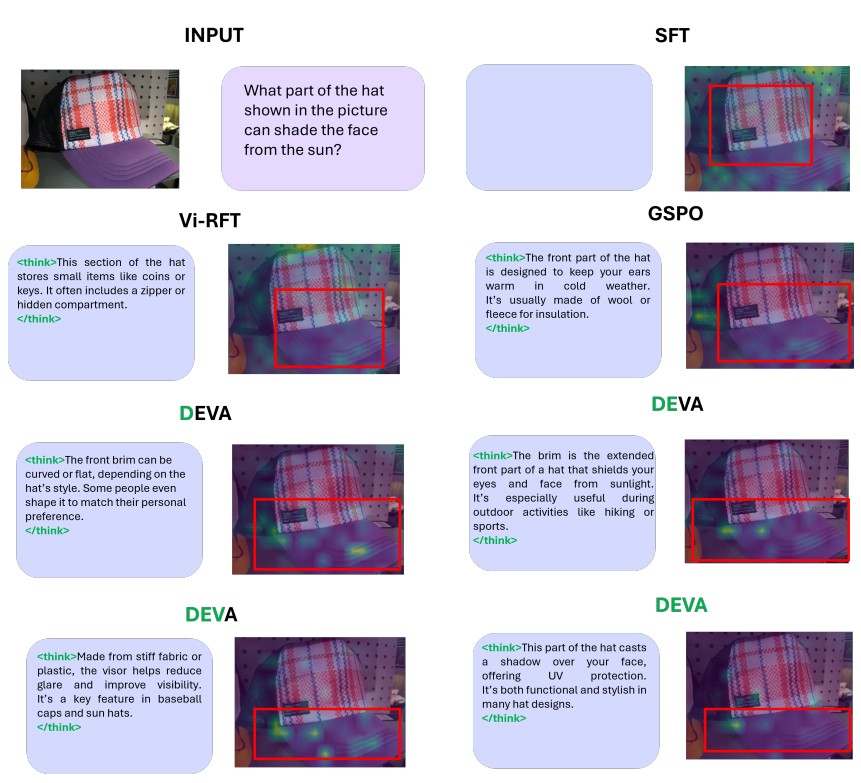

Figure 18: Visualization of the Hat as part of the LISA reasoning grounding dataset.

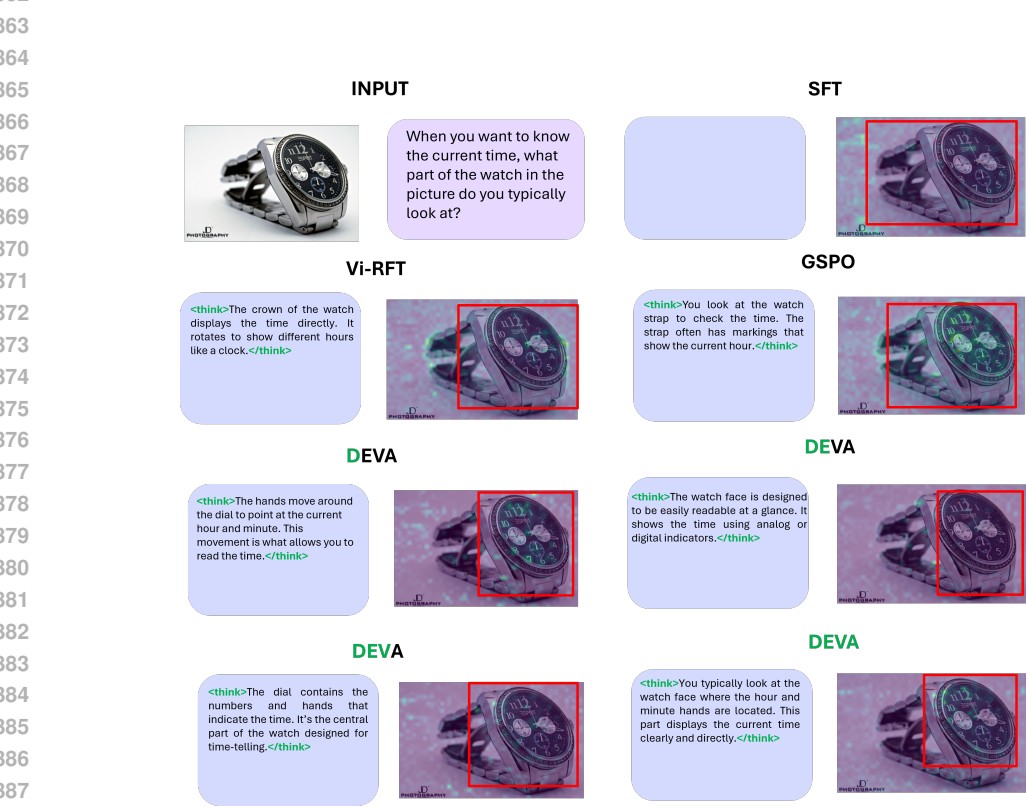

Figure 19:  Visualization of the Watch as part of the LISA reasoning grounding dataset.