# OpenReview forum: "DEVA: Fine-tuning Multimodal Large Language Models for Visual Perception Tasks"
_ICLR.cc/2026/Conference — ICLR 2026 Conference Withdrawn Submission_

### Official Review · Reviewer_WNa4 · 2025-10-30

**Soundness:** 2
**Presentation:** 2
**Contribution:** 2
**Rating:** 4
**Confidence:** 4

**Summary:**

This paper proposes DEVA, a framework to improve the reinforcement learning (RL) fine-tuning of Multimodal Large Language Models (MLLMs) for visual perception tasks. It identifies and addresses key limitations in existing methods like Group Relative Policy Optimization (GRPO), including low generation diversity, constrained policy exploration, and suboptimal reward design. DEVA introduces a GFlowNet-based loss for Diversity, global entropic regularization for Exploration, an alignment Volume-based reward, and harmonic Aggregation. Experiments on the ViRFT benchmark, using Qwen2-VL models, show that DEVA significantly outperforms standard RL baselines on reasoning grounding and few-shot detection/classification tasks.

**Strengths:**

1. Novel and Comprehensive Framework: The paper presents a well-motivated framework that simultaneously addresses multiple, distinct limitations (diversity, exploration, reward formulation) of existing RL-based MLLM fine-tuning, rather than offering just a single, minor tweak.

2. Strong Empirical Results: DEVA demonstrates significant and consistent quantitative improvements over a wide range of recent RL baselines (including Visual-RFT, GSPO, and PAPO). The gains reported on the LISA reasoning grounding task (e.g., +5-13 mIoU points over Visual-RFT) are substantial.

3. Thorough Ablation and Analysis: The authors provide detailed ablation studies that effectively isolate and validate the positive contribution of each of DEVA's four components. Furthermore, the qualitative attention visualizations (Fig. 6a) offer valuable intuition, suggesting DEVA helps the model focus more precisely on relevant object regions.

**Weaknesses:**

1. Limited Model Baselines: The experiments are primarily conducted on Qwen2-VL (2B and 7B) and LLaVA 1.5. To better contextualize DEVA's performance against the current state-of-the-art, the comparison should be expanded to include more recent and powerful open-source MLLMs, such as models from the Qwen2.5-VL or Qwen3-VL series.

2. Limited Comparison Scope: While the paper compares against many RL-based fine-tuning methods, the evaluation landscape feels somewhat insular. It is unclear how a DEVA-tuned model would perform against other state-of-the-art models on these benchmarks (e.g., LISA, Perception-R1, ...) that may use different fine-tuning and RL strategies.

3. Computational Complexity: The DEVA framework introduces several new components, including a GFlowNet objective and an alignment volume calculation that requires an external feature extractor (BLIP-2). The paper lacks a discussion on the computational overhead (e.g., increased training time, GPU memory) of this added complexity compared to the vanilla GRPO (Visual-RFT) baseline.

4. External Dependency: The proposed alignment volume reward, a key contribution, relies on an external, pre-trained model (BLIP-2) to extract features. This introduces a dependency, and the framework's performance may be sensitive to the choice of this specific extractor, though this is briefly explored in the appendix.

**Questions:**

1. How well does the DEVA framework generalize when applied to more powerful and recent base MLLMs, such as Qwen2.5-VL or Llama 3-V? Do the significant relative gains persist?

2. Could the authors please quantify the computational overhead (e.g., additional training time and/or memory usage) introduced by the full DEVA framework compared to the Visual-RFT baseline?

3. The alignment volume reward is a novel contribution. How sensitive is its performance to the choice of the external feature extractor (BLIP-2)? Have the authors experimented with using the MLLM's own vision encoder embeddings to calculate this volume, which would remove the external dependency?

4. The GFlowNet diversity loss (Eq. 3) uses a specific reward function based on the reference model $\pi_{ref}$. What is the intuition behind this particular reward design, and how sensitive is the model's performance to the $\gamma$ hyperparameter?

5. The framework is evaluated on visual perception tasks. Have the authors considered applying DEVA to other common MLLM tasks, such as genral perception (detection, counting, and OCR), general-purpose VQA or instruction-following, to assess if the benefits of enhanced diversity and exploration are more broadly applicable?

---

### Official Review · Reviewer_Acqc · 2025-10-30

**Soundness:** 2
**Presentation:** 2
**Contribution:** 2
**Rating:** 2
**Confidence:** 4

**Summary:**

This paper presents DEVA, which uses a combination of GFlowNet objective, adding global entropic divergence loss and a non-verifiable reward that minimizes hypervolume among different representations, all of which achieve help achieve improvements in different metrics and training dynamics.

**Strengths:**

The paper presents improvements in terms of performance and training dynamics, and conducts ablations highlighting the improvements of each of the building blocks of DEVA.

**Weaknesses:**

- The paper compares against RL baselines but doesnt really compare against other algorithmic approaches for purposes of improving diversity in responses which is more relevant to the motivations of this work. One such a comparison is to optimize other objectives (e.g. pass@k) which has been empirically shown to help improve diversity in responses. There are possibly other lines of work that consider diversity of responses (that I maybe unaware of).
- In general, the paper presents a mix of approaches which appear to be barely related (particularly relating to minimizing hyper-volume).

**Questions:**

Please refer to the weaknesses section.

---

### Official Review · Reviewer_Xr48 · 2025-11-01

**Soundness:** 2
**Presentation:** 3
**Contribution:** 2
**Rating:** 4
**Confidence:** 4

**Summary:**

This paper introduces DEVA, a multi-component optimization framework designed to improve the Reinforcement Learning (RL) fine-tuning of Multimodal Large Language Models (MLLMs). The work identifies key limitations in prior methods, namely low sample diversity and suboptimal reward functions. DEVA integrates a suite of four techniques to address these issues: (1) GFlowNet for generating diverse candidate responses, (2) a novel "Alignment Volume" reward metric based on the geometric relationship between multimodal embeddings, (3) sequence-level entropy regularization to encourage exploration, and (4) a harmonic mean to balance multiple reward signals. Experiments conducted on models such as Qwen2-VL demonstrate that DEVA achieves significant performance improvements over baselines on several visual perception benchmarks.

**Strengths:**

1. **Synergistic Multi-Component Design:** The framework combines several targeted solutions into a cohesive and effective system. It integrates GFlowNet to enhance reward diversity, a novel geometric reward ("Alignment Volume") for better cross-modal alignment, sequence-level regularization for greater exploration freedom, and a robust reward aggregation method. Together, these components create a more stable and meaningful learning signal than prior methods.
2. **Strong Empirical Support:** The framework's effectiveness is demonstrated through rigorous experimentation. Thorough ablation studies clearly quantify the contribution of each individual component, and the results show consistent performance gains over various RL baselines and demonstrate generalization across different MLLM architectures.
3. **Enhanced Interpretability and Diagnostics:** The framework provides greater insight into the model's decision-making process. The "Alignment Volume" reward is inherently more interpretable than a simple scalar score, and its specificity is further improved by using decoder attention to focus only on question-relevant image patches. Additionally, visualizations like attention heatmaps provide direct, qualitative evidence that the model learns to better focus on target regions, offering a clear diagnostic tool to understand its behavior.

**Weaknesses:**

1. **Underdeveloped Theoretical Basis for "Alignment Volume":** The framework's core contribution, the "Alignment Volume," lacks a rigorous theoretical justification. The paper does not fully explore why minimizing this volume is an optimal proxy for cross-modal alignment, nor does it address the potential risk of representational collapse, where the model might produce semantically poor outputs that still satisfy the geometric objective. The method's performance is also tightly coupled to an external, pre-trained BLIP-2 model, and the sensitivity of the results to this specific architectural choice is not analyzed.
2. **Unverified Generalization Across Modalities and Tasks:** The framework's effectiveness has only been demonstrated in a narrow context. Its applicability remains unproven in several key areas:
    - **More Complex Visual Tasks:** The evaluation is limited to specific perception tasks, omitting more complex challenges like compositional reasoning or multi-turn visual dialogue.
    - **Multi-Image and Video Inputs:** The current formulation of the reward is designed for single image-text pairs, and its extension to scenarios with multiple images or video frames is not discussed.
    - **Text-Only Domains:** The paper does not investigate whether the non-visual components of the framework could provide benefits in purely textual fine-tuning tasks, which would help to disentangle the general RL improvements from the vision-specific ones.

**Questions:**

1. Could you provide a more rigorous theoretical justification for the "Alignment Volume" as a reward signal? How does this formulation mitigate the potential risk of representational collapse, where semantic diversity might be sacrificed for a lower volume score? Furthermore, how sensitive are the results to the choice of the external feature extractor?
2. How do you expect the performance gains from DEVA to translate to more complex visual tasks that require deeper reasoning, such as compositional understanding or multi-turn interaction?
3. What are the primary challenges in adapting the "Alignment Volume" concept to inputs involving multiple images or video? Would the core formulation need to be fundamentally changed?
4. Have you considered evaluating the non-visual components of DEVA in a text-only scenario? This could help clarify how much of the framework's benefit stems from general improvements to RL fine-tuning versus modality-specific alignment.

---

### Official Review · Reviewer_77cN · 2025-11-01

**Soundness:** 3
**Presentation:** 3
**Contribution:** 3
**Rating:** 4
**Confidence:** 4

**Summary:**

This paper proposes a framework (DEVA) to improve GRPO for adapting MLLMs to visual perception tasks. The proposed DEVA enhances diversity via a flow-based training objective, encourages broader policy exploration through global entropic regularization, and leverages alignment volume as a non-verifiable reward combined with harmonic aggregation. Extensive evaluations on few-shot classification, few-shot object detection, and reasoning grounding tasks using Qwen2-VL demonstrate the performance superiority over Visual-RFT baselines and recent RL variants (i.e., PAPO, DAPO, GSPO).

**Strengths:**

- **Extensive evaluation and significant performance:** DEVA is evaluated across multiple datasets, tasks, and models, showing clear and consistent gains over many RL-based baselines such as GRPO, PAPO, GSPO, and BNPO. Its improvements over the baseline and advantages over other RL-based methods are impressive.

- This paper provides a well-motivated and systematic extension to GRPO by jointly addressing the issues of diversity, exploration, and reward aggregation.

**Weaknesses:**

- **Incremental techniques:** DEVA is a practical framework that systematically combines existing techniques, such as GFlowNet losses, entropy regularization, and harmonic aggregation, to improve GRPO-based visual reinforcement fine-tuning. The framework is useful but offers limited novelty.

- **Justification of the proposed non-verifiable reward:** The paper acknowledges that alignment hyper-volume is non-verifiable (lacking ground truth). However, without verification, how do we know this reward aligns with task performance? Figure 2 shows reward dynamics, but this doesn't validate that minimizing volume indeed improves grounding performance. For grounding, the true objective is IoU or pixel-level accuracy. The alignment volume minimizes geometric coherence between image, query, and response embeddings. They are not the same. Why can this reward alone enhance grounding?

**Questions:**

1. Can you provide a formal analysis or intuition for why minimizing the hyper-volume among image, query, and response representations (i.e., Eq. 6)? The connection between geometric coherence of embeddings and task success seems indirect.

---

### Note · Authors · 2025-11-12

I have read and agree with the venue's withdrawal policy on behalf of myself and my co-authors.